# L2RSI: Cross-view LiDAR-based Place Recognition for Large-scale Urban Scenes via Remote Sensing Imagery

**Ziwei Shi**[1,2] **Xiaoran Zhang**[1,2] **Wenjing Xu**[1,2] **Yan Xia**[3]
**Yu Zang**[1,2*] **Siqi Shen**[1,2] **Cheng Wang**[1,2]

[1] Fujian Key Laboratory of Sensing and Computing for Smart Cities, Xiamen University, China
[2] Key Laboratory of Multimedia Trusted Perception and Efficient Computing,
Ministry of Education of China, Xiamen University, China
[3] University of Science and Technology of China, China
shizw1995@stu.xmu.edu.cn    zangyu7@xmu.edu.cn

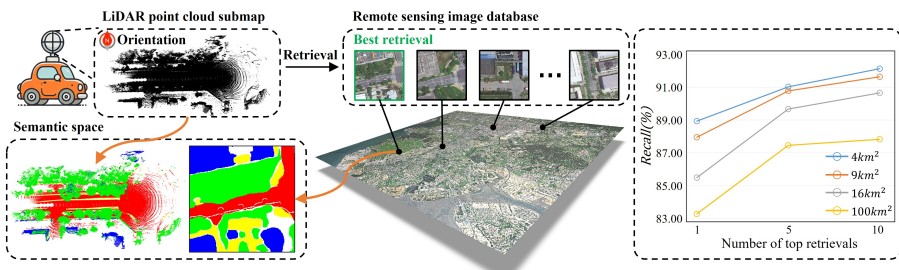

Figure 1: (Left) We propose L2RSI for cross-view LiDAR-based place recognition without 3D pre-built map. Given a point cloud submap representing the surroundings of the robot and the absolute orientation provided by a LiDAR and a magnetometer, L2RSI provides the most probable location in a large-scale city using high-resolution remote sensing imagery. (Right) Place recognition performance in different retrieval ranges. Notably, L2RSI achieved a recall ($< 30m$) of over $80\%$ at Top-1 retrieval within a range of $100km^2$.

## Abstract

We tackle the challenge of LiDAR-based place recognition, which traditionally depends on costly and time-consuming prior 3D maps. To overcome this, we first construct LiRSI-XA dataset, which encompasses approximately $110,000$ remote sensing submaps and $13,000$ LiDAR point cloud submaps captured in urban scenes, and propose a novel method, L2RSI, for cross-view LiDAR place recognition using high-resolution Remote Sensing Imagery. This approach enables large-scale localization capabilities at a reduced cost by leveraging readily available overhead images as map proxies. L2RSI addresses the dual challenges of cross-view and cross-modal place recognition by learning feature alignment between point cloud submaps and remote sensing submaps in the semantic domain. Additionally, we introduce a novel probability propagation method based on particle estimation to refine position predictions, effectively leveraging temporal and spatial information. This approach enables large-scale retrieval and cross-scene generalization without fine-tuning. Extensive experiments on LiRSI-XA demonstrate that, within a $100km^2$ retrieval range, L2RSI accurately localizes $83.27\%$ of point cloud submaps within a $30m$ radius for top-1 retrieved location. Our project page is publicly available at https://shizw695.github.io/L2RSI/.

*Corresponding Author

39th Conference on Neural Information Processing Systems (NeurIPS 2025).

# 1 Introduction

Place recognition aims to retrieve the closest match and its location from a pre-built database within the global coordinate system when GPS is weak or even denied, serving as an essential task for autonomous driving and robotic navigation. LiDAR-based place recognition is now increasingly attractive since 3D point clouds from LiDAR are invariant to lighting and shadows [1].

Existing LiDAR-based place recognition methods face a significant limitation: they depend on up-to-date prior 3D maps, the acquisition and maintenance of which are time-consuming and costly. Recent research has sought to address this issue by exploring alternative approaches. Tang et al. [2, 3] unified overhead imagery and LiDAR into a pseudo point cloud modality, achieving retrieval-based place recognition. However, their approach only works on the assumption of a known route, which confines the retrieval process to a very limited range. Cho et al. [4] conducted LiDAR-based place recognition using OpenStreetMap (OSM). Due to the dual challenges of cross-view and cross-modality, existing methods typically restrict the retrieval area to a small range. The most commonly used 3D datasets include the Oxford Radar RobotCar [5], KITTI [6] and KITTI-360 [7]. The former consists of multiple repeated trajectories, covering an area of less than $2km^2$, while the latter two datasets are composed of several independent trajectories, with the largest covering range approximately $1km^2$.

Table 1: List of conditions and tasks.

| Conditions | |
|---|---|
| GPS | ✗ |
| IMU | ✗ |
| Query | LiDAR+MAG |
| Reference | RSI |
| Tasks | |
| Range | $100km^2$ |
| Database | >60000 |
| Distance | <30m |
| $Recall@1$ | >80% |

As shown in Table 1, our task is to utilize high-resolution remote sensing imagery (RSI) as the database (due to the advantages of global coverage, cost-effectiveness, and timeliness), to achieve place recognition (LPR) in the range of over $100km^2$ urban scenes, more than $60000$ RSI and $30m$ required distance threshold, based on the only sensors of LiDAR and magnetometer (MAG).

In fact, using high-resolution remote sensing as the reference map will lead to a large gap between the query and databases in terms of domain and view, which will cause two major problems: (1) Most of the current retrieval solutions can hardly learn the correspondence between the query point cloud and the remote sensing image retrieval database directly. (2) Single query can be ambiguous and unstable, which limit the practicality of the system, not to mention the retrieving range of over $100km^2$.

To address the first problem, we observe that the content and style between LiDAR point clouds and remote sensing imagery, even from the same location, can differ significantly. However, they exhibit strong correlations in the semantic domain, as illustrated in the bottom-left corner of Fig. 1. Inspired by this observation, we propose a framework, named L2RSI, which includes a semantic contrastive learning network to align the global descriptors of LiDAR point cloud submaps and remote sensing submaps in the semantic space. The point cloud submaps are obtained through the registration of short-range LiDAR scans, overcoming the sparsity of single-frame point clouds. With the direction provided by an additional magnetometer (MAG), these submaps are transformed into a bird's-eye view (BEV) representation through rotation and projection, enabling the network to effectively exploit the shared semantic features visible in both modalities.

Specific to the second problem, we propose the Spatial-Temporal Particle Estimation (STPE) to aggregate spatio-temporal information of continuous queries, thus to eliminate the ambiguity between geometrically distant but semantically similar positions. Specifically, the retrieval results in each query are considered as particles. Instead of filtering out the incorrect particles, our idea is to aggregate these particles and estimate a more accurate probability density distribution for the current position. Distribution of the particles of a single query can be modeled as the mixture of multiple gaussian models (GMMs). Then, the aggregation spread over a desired time window through an estimated inter-frame correspondence of the point clouds, estimating the probability density of the current query and refining the retrieval results.

Our contributions are summarized as follows:

- To the best of our knowledge, the proposed framework is the first to address the challenge of large-scale (over $100km^2$) urban cross-view LiDAR-based place recognition with high-resolution remote sensing imagery, demonstrating promising performance with significant practical applicability.

- We propose a particle estimation algorithm that utilizes the mixture of multiple gaussian models to aggregate spatio-temporal information and infer the probability density of the current position, significantly improving the retrieval performance.

- The proposed method exhibits remarkable generalization ability, achieves $Recall@1 (< 30m)$ of $83.27\%$ within a retrieval range of $100km^2$, and demonstrates promising potential on cross scene generalization without fine-tuning.

## 2 Related work

**Uni-modal 3D place recognition.** The early work by Uy et al. proposed PointNetVLAD [8], which utilized PointNet [9] as a point cloud feature extractor, followed by a NetVLAD layer [10] for global feature aggregation. LPD-Net [11] additionally introduced local information of points and aggregated global descriptors through GNN. SOE-Net [1] incorporated orientation encoding into the feature embedding process to capture spatial context and used a self-attention mechanism to aggregate global information. MinkLoc3D [12] applied sparse convolution and Generalized-Mean pooling [13] to extract discriminative features from sparse voxel representations of point clouds, and its extension, MinkLoc++ [14], further incorporated monocular camera images. SVT-Net [15] extended the transformer network to sparse voxels for capturing long-range contextual features. Zhou et al. [16] encoded geometric information by representing point clouds as normal distribution transformation cells. CASSPR [17] leveraged a dual-branch hierarchical cross-attention mechanism to integrate the advantages of point-based and voxel-based features. Crossloc3d [18] proposed a ground LiDAR place recognition method using aerial LiDAR as the database. However, such methods require an available and reliable prior 3D map, which is a demanding requirement in practice.

**Cross-modal 3D place recognition.** Cross-modal methods enhance usability by replacing costly prior 3D maps with data from alternative modalities. The work by Lee et al. [19] was the first to achieve LiDAR-based place recognition within a street-view image database. They created depth images from short sequence images using MantDepth [20] and established a consistent data representation with LiDAR range images through contrastive learning. Lip-Loc [21] aligned LiDAR point clouds with RGB camera images through range image projection. C2L-PR [22] achieved modality alignment by 2D to 3D mapping via depth estimation and semantic segmentation, and mitigates the FOV discrepancy through orientation voting. VXP [23] incorporated similarity constraints on local descriptors and established a voxel-pixel shared feature space through a two-stage training process. Text2Pos [24] and Text2Loc [25] determined the most probable location using textual descriptions. UniLoc [26] designed a universal place recognition working with any single query modality. Some researchers focused on publicly available maps, such as OpenStreetMap (OSM). Ruchti et al. [27] relied exclusively on road information extracted from LiDAR point clouds, associating it with the OSM road network to compute weights for Monte Carlo localization. Cho et al. [4] designed a hand-crafted rotation-invariant descriptor to retrieve the most similar location from the OSM database based on the relative position of LiDAR and buildings. The works most relevant to our research are a series of studies based on overhead images. Tang et al. [2] utilized ray-tracing to generate pseudo-point clouds from LiDAR scans and overhead images that are more similar to each other. In subsequent work [3], they introduced CycleGAN [28] to generate synthetic LiDAR images from satellite images and selected pseudo-pairs from sequential data to learn a shared embedding space. The place recognition range of the aforementioned methods is limited to known or predefined routes within urban scenes. In contrast, our L2RSI can be easily extended to unknown areas spanning tens to hundreds of square kilometers, offering broader application potential.

**Sequence-Constrained Place Recognition.** Garg et al. [29] and Malone et al. [30] introduced temporal information by training a sequence-matching network to enhance place recognition performance. However, remote sensing imagery is inherently static and lacks temporal continuity or directional cues, and such methods are not applicable. Prticle filtering [27, 31] is widely used in localization and tracking tasks, where it recursively refines the particles based on the motion model of the system, gradually converging them towards the position of the robot. The main difference from the proposed STPE is that our approach is particle aggregation rather than filtering. This is due to the inherent content discrepancies between remote sensing imagery and LiDAR data, where the observation information from a single query is often highly unreliable. Such toxic queries can lead to the filtering out of most, if not all, trustworthy particles. In contrast, the proposed method ensures the smooth transmission of spatio-temporal information and is more suitable for extremely challenging place recognition.

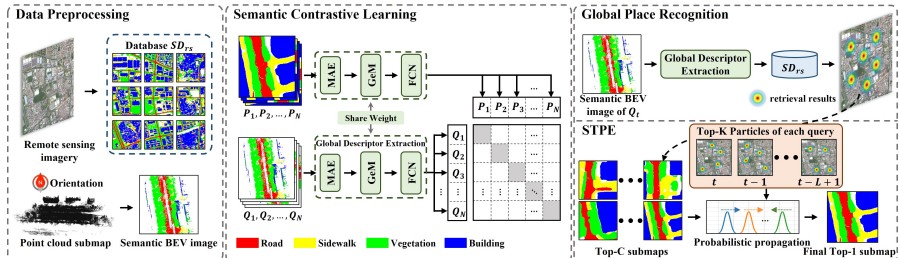

Figure 2: Overview of the proposed L2RSI. It consists of three modules: the data preprocessing module (left), the training framework for Global Descriptor Extraction network (middle) and the inference framework (right).

**Cross-view 2D place recognition.** Since our method unifies cross-modal data into semantic images, some image-based place recognition methods can also be applied. Early works such as CVUSA [32], CVACT [33], and VIGOR [34] established the main benchmarks in image place recognition between street view and satellite view. GeoDTR [35], used CNNs to extract global features and enforced the introduction of a Transformer to extract additional information through a counterfactual loss. Sample4Geo [36] introduced the off-the-shelf ConvNeXt-B as a feature extractor and designed a dynamic similarity sampling strategy to construct training batches. Recent FRGeo [37] proposes a feature reorganization module to effectively enhance feature aggregation for BEV images. In this work, we find that directly applying these methods can not work well in our task due to the domain gaps.

## 3 Problem statement

We begin by defining the large-scale remote sensing imagery $M_{ref} = \{m_i\}_{i=1}^{M}$ to be a collection of square submaps $m_i$. Each submap is sampled at equal intervals with partial overlap and includes the UTM coordinates of the center point. Let $Q$ be a query point cloud submap obtained by aggregating a short sequence of LiDAR scans. Our task is defined as determining the coarse position of the center point in the query point cloud submap, which is a cross-view, cross-modal retrieval problem:

$$m^* = \underset{m_i \in M_{ref}}{argmax} \, Sim(F_q(Q, o), F_{ref}(m_i)), \tag{1}$$

where $F_q(\cdot)$ and $F_{ref}(\cdot)$ denote the deep learning networks used to encode the point cloud submap $Q$ and remote sensing submaps $m_i$ into a shared feature space. $Sim(\cdot, \cdot)$ represents a measure of similarity, *e.g.*, the dot product. $o$ represents the orientation of the intermediate frame in $Q$. We obtain a noisy measurement $o$ from a magnetometer.

## 4 Methodology

Fig. 2 shows the overview of the proposed L2RSI. In the Data preprocessing module (Sec. 4.1), on one hand, we perform semantic segmentation on remote sensing imagery, which are then evenly divided into submaps to serve as the semantic database. On the other hand, we conduct semantic segmentation on point cloud submaps, which are subsequently compressed into BEV images. During training, we learn representations of BEV images and remote sensing submaps within a shared semantic space to overcome the modality gap (Sec. 4.2). During inference, we extract the global descriptor of the given query point cloud submap and retrieve a set of potential candidates from the semantic database. Then we apply STPE to estimate the probability density of the current query position and refine these candidates (Sec. 4.3).

### 4.1 Data preprocessing

**Remote sensing imagery.** Satellite and LiDAR data exhibit discrepancies in content and style due to differences in viewpoints, imaging principles, and acquisition times. In practice, humans typically assess whether the point cloud and remote sensing imagery represent the same location by identifying high-level semantic information. Inspired by this, we employ AIE-SEG developed by Alibaba[1], which is a well-established universal segmentation model for remote sensing interpretation, to perform semantic segmentation on remote sensing imagery. Intuitively, we select roads, sidewalks,

---

[1] https://engine-aiearth.aliyun.com/#/portal/analysis

vegetation, and buildings as the semantics of interest. Subsequently, a square window with side length $d$ slides uniformly across the semantic image, extracting submaps to form the semantic database, as shown in the top-left corner of Fig. 2. Notably, considering that robots travel along the road, we filter out submaps that are located far from the road label.

**Point cloud.** For the acquired short sequences of LiDAR scans, we build the point cloud submap using FastGICP [38]. Then, we introduce SphereFormer [39] to perform segmentation based on the same semantics as remote sensing imagery. To adapt to data collected by different LiDAR systems, we remove the intensity feature and retrain the model on SemanticKITTI [40]. While various unsupervised domain adaptation methods can potentially yield better semantic segmentation results, our approach is simple yet effective.

To obtain the same perspective as the remote sensing imagery, we then compress the point cloud into a bird's-eye view (BEV) image, covering an area of approximately $d \times d$. The topmost points are retained to preserve the features that are commonly visible in the remote sensing imagery. Based on the orientation measured by the magnetometer, we rotate the BEV image to align the north direction upwards before feeding it into the global descriptor extraction network.

## 4.2 Semantic contrastive learning

Recently, several works focused on building cross-modal representations have emerged prominently across various zero-shot computer vision tasks, such as classification [41], recognition [42, 43, 44] and retrieval [45, 46]. Inspired by their work, we design a network that aligns cross-modal semantic features in a unified feature space through contrastive learning.

**Network architecture.** As shown the middle of Fig. 2, we employ a dual-branch network to extract global descriptor for pairs of semantic images $\{Q_i\}_{i=1}^N$ from point clouds and $\{P_i\}_{i=1}^N$ from remote sensing imagery, where $N$ is the batch size. The distance between the centers of the paired images is constrained to be less than $d/2$. Each branch consists of a semantic encoder, a generalized mean (GeM) pooling, and a fully connected (FCN) layer in a sequential configuration. The semantic encoder supports initialization with various pre-trained foundation models. We use MAE [47] here. Then, GeM pooling and the FCN layer generate global descriptors. Since both branches operate within the semantic domain, we fully share their weights. In this work, we set $d = 60m$.

**Loss function.** During training, we ensure that image pairs within a batch serve as negatives for each other and use the symmetric InfoNCE loss as the contrastive learning objective [36], formulated as follows:

$$\mathcal{L} = -log(\frac{exp(f_i^Q \cdot f_i^P/\tau)}{\sum_{j \in N} exp(f_i^Q \cdot f_j^P/\tau)}) - log(\frac{exp(f_i^P \cdot f_i^Q/\tau)}{\sum_{j \in N} exp(f_i^P \cdot f_j^Q/\tau)}), \tag{2}$$

where $f_i^Q$ and $f_i^P$ represent the global descriptors of $Q_i$ and $P_i$, respectively. The temperature coefficient $\tau$ is set to 0.1. Compared to the traditional triplet loss [8, 2, 3], the advantage is that increasing the number of negatives in a training batch greatly accelerates the training process.

## 4.3 Spatial-Temporal Particle Estimation

During inference, we first extract the global descriptor of the query point cloud submap $Q_t$ at time step $t$. We then calculate its similarity distance to the image submaps in the remote sensing database for determining the Top-C retrieval results. ( see the top-right corner of Fig. 2). Note that the global descriptors are extracted offline to improve retrieval efficiency.

To alleviate the unreliability of individual query retrieval results, we consider the query sequence $\{Q_j\}_{t-L+1}^t$ from time step $t-L+1$ to $t$. The retrieval results of $Q_j$ are treated as particles, where the state of each particle, denoted as $\mathbf{x}_j^{[k]}$ with $k \in 1, 2, ..., K$, refers to the 2D coordinates of candidate submaps in the remote sensing database. The state of the $k$-th particle of $Q_j$ is transitioned from time step $t-1$ to $t$ through the following function:

$$\mathbf{x}_t^{[k,j]} = M_{t-1}(\mathbf{x}_{t-1}^{[k,j]}) + \mathbf{v}_t^{[k,j]}, \tag{3}$$

where $M_{t-1}$ is the motion model from time step $t-1$ to $t$ and $\mathbf{v}_t^{[k,j]}$ a noise term. Unlike traditional particle filtering, we propagate particles from multiple past time steps to current time step $t$ using the motion model, and then construct a probability density function at time step $t$ based on the combined particle distribution.

In practice, we obtain the relative positions between LiDAR scans from FastGICP [38], and correct the direction in the world coordinate system every 20 meters using the magnetometer, which subsequently enables the determination of particle states at each time step. We observe that these particles at each time step tend to cluster around some locations with similar features, which can be formulated as multiple Gaussian distributions:

$$P(x,y) = \sum_{m=1}^{M} A_m \cdot \exp\left(-\frac{(x-\mu_{xm})^2}{2\sigma_{xm}^2} - \frac{(y-\mu_{ym})^2}{2\sigma_{ym}^2}\right), \tag{4}$$

where $M$ represents the number of Gaussian distributions and we use DBSCAN clustering top-K candidates ($K < C$) to determine it with a radius of $r = 30m$. Let $x$ and $y$ denote the coordinates of the particles. Since they are assumed to be independent, their covariance is 0. $A_m$ indicates the contribution of the $m$-th Gaussian distribution to the overall distribution, which is computed as follows:

$$A_m = \frac{N_m}{\sum_{i=1}^{M} N_i}, \tag{5}$$

where $N_i$ is the number of candidates in the $i$-th cluster. The candidates in $m$-th cluster are represented as $\{(x_i^m, y_i^m)\}_i^{N_m}$. The center of Gaussian distribution $\mu_{xm}$ and $\mu_{ym}$ is the geometric center of $m$-th cluster:

$$(\mu_{xm}, \mu_{ym}) = \left(\frac{1}{N_m}\sum_{i=1}^{N_m} x_i^m, \frac{1}{N_m}\sum_{i=1}^{N_m} y_i^m\right), \tag{6}$$

The standard deviations $\sigma_{xm}$ and $\sigma_{ym}$ are estimated as follows:

$$\sigma_{xm} = \sqrt{\frac{1}{N_m}\sum_{i=1}^{N_m}(x_i^m - \mu_{xm})^2}, \sigma_{ym} = \sqrt{\frac{1}{N_m}\sum_{i=1}^{N_m}(y_i^m - \mu_{ym})^2}, \tag{7}$$

Therefore, we estimate the probability density function of the query $Q_t$ by propagating the retrieval result distribution of the query sequence $\{Q_j\}_{t-L+1}^t$ from time step $t-L+1$ to $t$. Assuming the relative position between $Q_j$ and $Q_t$ is $(\Delta x_j, \Delta y_j)$. The distribution of the retrieval results of $Q_j$ after propagation according to (4) is:

$$\tilde{P}_j(x,y) = \sum_{m=1}^{M} A_m \cdot \exp\left(-\frac{(x-\mu_{xm}^j)^2}{2\sigma_{xm}^2} - \frac{(y-\mu_{ym}^j)^2}{2\sigma_{ym}^2}\right), \tag{8}$$

where $\mu_{xm}^j = \mu_{xm} + \Delta x_j$ and $\mu_{ym}^j = \mu_{ym} + \Delta y_j$. This implies that the particle states at the current time step are derived from their estimated values at the previous step, rather than from the positions associated with the retrieved remote sensing images. The probability density function of the position at time step $t$ can be formulated as a Gaussian Mixture Model:

$$P_t(x,y) = \frac{1}{L}\sum_{j=t-L+1}^{t} \tilde{P}_j(x,y), \tag{9}$$

where the parameters of $\tilde{P}_j(x,y)$ is updated as the vehicle moves. Then, we assign a probability value to each remote sensing submaps in the database:

$$P_t(u,v) = \frac{1}{S}\int_{u-r}^{u+r}\int_{v-r}^{v+r} P_t(x,y)\,dy\,dx, \tag{10}$$

where $(u,v)$ is the center coordinate of the remote sensing submap. We take the average value within a rectangle with an area of $S = 4r^2$ as the probability value for this remote sensing submap. To avoid cumulative errors, we only consider the query sequence of $250m$, and re-rank the retrieval results of $Q_t$ based on probability values.

## 5 Experiment

### 5.1 Benchmark dataset

Due to the lack of publicly available datasets that provide correspondences between large-scale outdoor LiDAR scans and remote sensing imagery, we construct the LiRSI-XA and LiRSI-Oxford datasets for training and evaluation.

**LiRSI-XA.** We purchase high-resolution remote sensing imagery covering Xiang'an District from AIRSAT Technology Group, with a resolution of $0.5m$. We extract submaps of $60m \times 60m$ from

Table 2: Summary statistics for dataset.

| Submap Num. | LiRSI-XA | | | | | LiRSI-Oxford | |
| --- | --- | --- | --- | --- | --- | --- | --- |
| | Training Set | Test Set-S | Test Set-M | Test Set-L | Test Set-G | 11-14-02-26 | 14-12-05-52 |
| PC | 12194 | 862 | 862 | 862 | 862 | 1697 | 1671 |
| RSI | 47913 | 3094 | 6479 | 12119 | 63047 | 2103 | 2103 |
| Area ($km^2$) | 65 | 4 | 9 | 16 | 100 | 2.8 | 2.8 |

the original remote sensing imagery at intervals of $20m$, resulting in a two-third overlap between adjacent submaps. And the submaps lacking road semantics in their $30m \times 30m$ central region are filtered out.

We utilize a Ouster OS1-64 LiDAR to acquire point cloud data in the Xiang'an District of Xiamen, covering a trajectory of approximately $100km$. We record directional information through a magnetometer with an average error of less than $10°$. During point cloud preprocessing, we construct a point cloud submap every $5m$ and compress it into a semantic BEV image with a side length of $d = 60m$. There is a three-year temporal gap between LiDAR data and remote sensing imagery.

The vehicle trajectory is shown in Fig. 3a. We select approximately $5km$ of the trajectory as the test set, with the remaining portion used for training. For the training set, we manually exclude data with inherent misalignment between ground and aerial semantic information to avoid confusion during training. This includes the areas where the sky is obstructed, such as the road segments beneath overpasses and dense street trees, as well as changes in buildings, vegetation, and roads due to the temporal gap. For the test set, we established databases of $4km^2$ (S), $9km^2$ (M), $16km^2$ (L) and $100km^2$ (G) to evaluate the retrieval performance of the proposed method, see Table 2. Notably, the point cloud at the boundary between the test and training sets are discarded, ensuring that the network is never exposed to the test set data during training.

**LiRSI-Oxford.** Oxford Radar RobotCar is a famous urban scene dataset [5] for localization. The data is collected with a Velodyne HDL-32E LiDAR. The dataset includes multiple repeated traversals, each covering about $10km$, as shown in Fig. 3b. Based on this, we supplement the database with the corresponding remote sensing imagery, covering an area of approximately $2.8km^2$. And we simulate the magnetometer by adding random noise within $\pm15°$ to the ground truth of orientation, as the official dataset does not include a magnetometer. The source and resolution of remote sensing imagery, as well as the data preprocessing process, are the same as those of the LiRSI-XA. The data of 11-14-02-26, 14-12-05-52 are used as the test set to evaluate the generalization ability of the proposed method without fine-tuning, see Table 2.

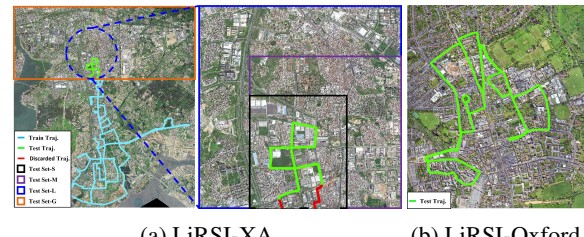

(a) LiRSI-XA  (b) LiRSI-Oxford

Figure 3: Dataset configuration. In LiRSI-XA a, different colored rectangles highlight test sets from databases of varying sizes, all sharing the same query trajectory (green). Besides, the training trajectory (light blue) and discarded trajectory at the boundary (red) are annotated. In LiRSI-Oxford b, the green trajectory is as the query for the test set, while the entire remote sensing imagery serves as the database.

## 5.2  Evaluation criteria

Following pervious place recognition task [8, 19], we use retrieval recall to evaluate LiDAR-based place recognition based on remote sensing imagery. $Recall@N$ means the recall at top-$N$ retrieval results, that is, the proportion of queries for which the correct result is retrieved within the top-$N$ results. We define a correct result as when the Euclidean distance between the center of the retrieved remote sensing submap and the center of the query point cloud submap is less than $30m$. We particularly emphasize on $Recall@1$, as it has the most direct practical relevance.

Table 3: Performance comparison on LiRSI-XA and LiRSI-Oxford. $Recall@1$ and $Recall@10$ (%) are reported. The best result is rendered in bold.

| Methods | $Recall@1$ / $Recall@10$ (%) ↑ | | | | | |
| --- | --- | --- | --- | --- | --- | --- |
| | Test Set-S | Test Set-M | Test Set-L | Test Set-G | 11-14-02-26 | 14-12-05-52 |
| LIP-Loc [21] | 16.82/53.48 | 13.69/45.36 | 11.60/41.65 | 11.02/39.44 | 0.88/10.78 | 0.96/11.13 |
| GeoDTR [35] | 8.68/25.27 | 6.25/19.36 | 4.74/14.95 | 4.39/9.26 | 0.47/3.18 | 0.48/3.71 |
| Sample4Geo [36] | 28.63/53.92 | 25.39/49.75 | 24.23/46.27 | 22.95/43.02 | 7.13/20.68 | 7.51/21.75 |
| FRGeo [37] | 22.26/49.88 | 19.72/42.45 | 17.16/38.05 | 15.54/34.45 | 1.62/7.34 | 1.58/7.86 |
| L2RSI (w/o STPE) | 30.05/68.56 | 23.90/61.95 | 21.93/57.42 | 20.07/52.55 | 10.19/30.23 | 11.25/30.58 |
| L2RSI (w. SuperGlobal [48]) | 30.39/57.54 | 23.90/53.02 | 22.27/48.26 | 20.19/42.92 | 8.9022.98 | 8.98/22.86 |
| L2RSI (w. PF) | 71.66/86.88 | 55.17/72.47 | 50.87/72.47 | 47.85/58.65 | 21.40/31.43 | 16.83/29.46 |
| L2RSI (w. STPE) | **88.93/92.13** | **87.95/91.64** | **85.49/90.65** | **83.27/87.82** | **42.29/59.41** | **43.77/62.76** |

## 5.3 Results

### 5.3.1 Evaluation on test sets

The proposed L2RSI is the first to perform cross-view LiDAR-based place recognition in large-scale urban scenes with high-resolution remote sensing imagery. Several state-of-the-art place recognition networks are capable of handling preprocessed cross-modal semantic images, such as cross-modal 3D place recognition (LIP-Loc [21]) and cross-view 2D place recognition (GeoDTR [35], Sample4Geo [36], and FRGeo [37]). Firstly, we benchmark them to evaluate the performance of our method in the test sets of LiRSI-XA with different size databases. The training settings on LiRSI-XA for all methods are consistent, except for the image size and descriptor dimensions, which follow the official configurations. Notably, we replace the LiDAR range image projection used in the original LIP-Loc with a BEV projection. The comparison is summarized in Table 3. L2RSI (w. STPE) significantly outperforms the baseline methods, reaches a $Recall@1$ of 88.93% with a database of $4km^2$. As the retrieval range expands, L2RSI (w. STPE) demonstrates increased robustness, with the database area increased to $100km^2$ and only a 5.66% degradation in $Recall@1$ and a 4.31% degradation in $Recall@10$. In addition, we also introduced a comparison of two refinement strategies, including SuperGlobal reranking [48] and classical particle filtering (PF). The former focuses on refining individual queries and database features, leading to only marginal improvements in performance. For the latter, we apply its principles within the retrieval framework, initializing the number of particles to 120. With the integration of sequence information, the performance of L2RSI (w. PF) shows considerable improvement. However, in challenging cross-view and cross-modal retrieval tasks, particles frequently decay, necessitating re-initialization and impeding the flow of sequence information. The proposed STPE substitutes filtering with aggregation, efficiently utilizing spatio-temporal information to substantially enhance the unstable performance of single query.

Secondly, we directly test the model on the LiRSI-Oxford without any fine-tuning. Given the stark differences in equipment models, road widths, architectural styles, etc., this challenge marks the highest level of generalization ability. In the lower part of Table 3, the baseline methods fail completely, while L2RSI (STPE) achieves $Recall@1$ over 40% and $Recall@10$ above 60% on both trajectories. Although the current results are not yet sufficient for real-world deployment, they demonstrate great potential for future applications. Please refer to Sec. A.3 for comparative experiments in more scenes.

Table 4: Ablation study about effectiveness of components. The results include $Recall@1$ and $Recall@30$ (%) on Test Set-S and 11-14-02-26. The best result is rendered in bold. **Sem.**: Semantic segmentation. **STPE**: Spatial-Temporal Particle Estimation. **Orient.**: Orientation.

| Sem. | STPE | Orient. | $Recall@1$ / $Recall@30$ (%) ↑ | |
| --- | --- | --- | --- | --- |
| | | | Test Set-S | 11-14-02-26 |
| | √ | √ | 50.43/67.90 | 8.13/25.42 |
| √ | | √ | 30.05/85.50 | 10.19/45.90 |
| √ | √ | | 54.37/76.14 | 4.92/21.24 |
| √ | √ | √ | **88.93/95.94** | **42.29/70.15** |

### 5.3.2 Ablation study

The following ablation studies evaluate the impact of different components and settings on the performance of L2RSI.

**Semantic information.** In the first row of Table 4, we bypass semantic images, directly extracting submaps from raw remote sensing imagery and projecting 3D coordinates to generate point cloud BEV images. Compared to the full L2RSI (last row), $Recall@1$ on test set-S decreases by 38.50%, and on LiRSI-Oxford, $Recall@1$ dropped to 8.13%. This demonstrates that semantics serve as

Table 5: Ablation study about the sampling rate $\lambda$. $Recall@1$ (%) on Test Set-S is reported. The runtime (ms) here only involves STPE. The best result is rendered in bold.

| Test set-S | 10% | 20% | 30% | 40% | 50% | 60% | 70% | 80% | 90% | 100% |
|---|---|---|---|---|---|---|---|---|---|---|
| $Recall@1$ (%) $\uparrow$ | 85.98 | 88.07 | **88.93** | 88.44 | 88.44 | 88.07 | 88.07 | 87.82 | 88.07 | 88.44 |
| $RT(ms)\downarrow$ | **24.1** | 29.4 | 31.7 | 34.7 | 38.0 | 40.2 | 44.3 | 47.5 | 52.2 | 55.4 |

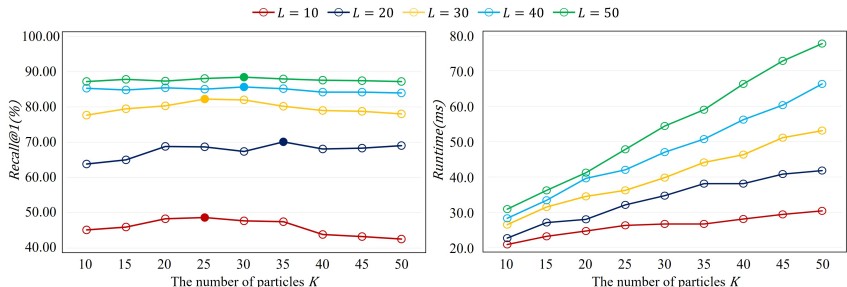

Figure 4: Ablation study about the number of queries in the sequence ($L$) and the number of particles for each query ($K$). Solid dots represent the optimal performance for each sequence length.

reliable information capable of bridging domain differences, playing a pivotal role in enhancing the generalization ability.

**STPE.** We ablate STPE in the second row of Table 4, and observe a performance drop of $58.88\%$ and $32.10\%$ in $Recall@1$ for the two test sets, respectively. In STPE, the GMMs are computed based on the top-K results ($K=30$) retrieved from the query sequence. Therefore, we also provide $Recall@30$ as a reference.

To further explore the impact of key parameters in STPE, we experiment with different settings for the number of queries $L$ in the sequence and the number of particles $K$ for each query. The results are presented in Fig. 4. We observe that $L$ is positively correlated with retrieval performance, while $K$ exhibits a peak. Additionally, larger values of $L$ and $K$ generally lead to an increased runtime. Considering the trade-off between performance and computational cost, $L = 50$ and $K = 30$ are adopted as default configurations.

In order to reduce the runtime, we introduce the sampling rate $\lambda$ to decrease the number of queries involved in STPE, without changing the sequence length. As shown in Table 5, when $\lambda = 30\%$, $Recall@1$ reaches $88.93\%$, while the refinement overhead per query is only $31.7ms$, demonstrating an excellent balance between performance and efficiency. This phenomenon is attributed to the fact that the adjacent point cloud submaps are spaced only $5m$ apart, resulting in highly redundancy. The sampled queries still encompass the majority of critical information. The qualitative results in Fig. 5 clearly illustrate the effectiveness of STPE.

### 5.3.3 Impact of Motion Model Failure

In STPE stage, dynamic objects such as pedestrians and vehicles may introduce disturbances into the motion estimation. To investigate the impact of progressive motion model degradation on place recognition performance, we introduce varying levels of yaw angle noise $\epsilon_{yaw}(°)$ and translational

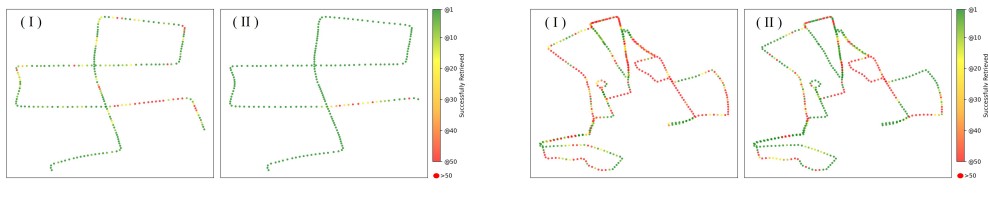

(a) retrieval results on Test Set-S          (b) retrieval results on 11-14-02-26

Figure 5: Visualization of retrieval results on LiRSI-XA a and LiRSI-Oxford b. Points closer to red indicate that more retrieval results are required to achieve correct retrieval at that location. (I) shows the results without STPE, and (II) shows the results with STPE.

Table 6: Impact of motion model failure in STPE stage. $Recall@1(\%)$ on Test Set-G are reported. The parentheses indicate the performance drop compared to the additional noise-free setting.

| $Recall@1(\%)$ | $\epsilon_{yaw} = \pm0°$ | $\epsilon_{yaw} = \pm5°$ | $\epsilon_{yaw} = \pm10°$ | $\epsilon_{yaw} = \pm15°$ | $\epsilon_{yaw} = \pm30°$ |
|---|---|---|---|---|---|
| $\epsilon_{xy} = \pm0$m | 83.27(-0.00) | 80.57(-2.70) | 78.72(-4.55) | 60.76(-22.51) | 48.46(-34.81) |
| $\epsilon_{xy} = \pm1$m | 82.77(-0.50) | 80.20(-3.07) | 77.85(-5.42)l | 59.77(-23.50) | 44.26(-39.01) |
| $\epsilon_{xy} = \pm4$m | 71.58(-11.69) | 67.04(-16.23) | 66.91(-16.36) | 56.70(-26.57) | 41.39(-41.88) |
| $\epsilon_{xy} = \pm9$m | 67.04(-16.23) | 64.94(-18.33) | 61.50(-21.77) | 54.37(-28.90) | 39.12(-44.51) |
| $\epsilon_{xy} = \pm16$m | 57.31(-25.96) | 53.87(-29.40) | 53.75(-29.52) | 50.92(-32.35) | 35.91(-47.36) |

noise $\epsilon_{xy}(m)$ into the estimated motion model. The translation noise is defined by the noise in the x-direction and y-direction as follows:

$$\epsilon_{xy} = \sqrt{(\epsilon_x^2 + \epsilon_y^2)}, \tag{11}$$

where we take $\epsilon_x = \epsilon_y$ for convenience. The average $Recall@1(\%)$ of three repeated experiments on Test Set-G ($100km^2$) of LiRSI-XA are shown in Table 6. Please note that the disturbances in the table represent intentionally introduced additional noise, based on the noise inherently caused by the FastGICP [38] algorithm and the magnetometer used in the system. $\epsilon_{yaw} = \pm5°$ and $\epsilon_{xy} = \pm1$m indicates that, in addition to the inherent noise, random yaw angle noise within a range of $\pm5°$ and random translation noise within a range of $\pm1m$ are introduced.

The results demonstrate that our method can tolerate a certain level of additional noise in the estimated motion model ($\epsilon_{yaw} < \pm10°$ and $\epsilon_{xy} < \pm1$m), which is typically introduced by dynamic objects affecting the FastGICP algorithm. In practice, we find that excluding the points of the data collection vehicle itself from the raw point cloud effectively reduces this noise to an acceptable level.

### 5.3.4 Computational cost analysis

Table 7: Computational cost analysis on Test Set-S. The number of parameters (M) and runtime (ms) are reported. The best result is rendered in bold.

| Metric | GDE | | STPE | |
|---|---|---|---|---|
| | P (M)↓ | $RT(ms)$↓ | P (M)↓ | $RT(ms)$↓ |
| LIP-Loc [21] | 46.09 | 5.7 | / | / |
| GeoDTR [35] | **26.82** | 6.5 | / | / |
| Sample4Geo [36] | 88.59 | 7.6 | / | / |
| FRGeo [37] | 27.82 | **4.6** | / | / |
| L2RSI | 86.58 | 5.8 | / | 31.7 |

In this section, we analyze the number of parameters and runtime of L2RSI in the global descriptor extraction and STPE stages. As reported in Table 7, since Sample4Geo and the proposed L2RSI use heavier backbones for feature extraction, *i.e.*, convnext-B and MAE-B, they have more parameters. However, the runtime difference during the global descriptor extraction (GDE) is minimal. STPE significantly refines retrieval results, with an additional cost of $31.7ms$. Nevertheless, the total time of $37.5ms$ still allows for real-time operation ($< 100ms$).

## 6 Conclusion

We present L2RSI, the first method for cross-view LiDAR-based place recognition within large-scale (over $100km^2$) urban scenes using high-resolution remote sensing imagery. We unify LiDAR point cloud submaps and remote sensing submaps into a shared semantic space via a carefully-designed contrastive learning network. Furthermore, we propose the Spatial-Temporal Particle Estimation to aggregate the spatio-temporal constraints of sequential queries, thereby further refining the performance of global place recognition. Extensive experiments on the LiRSI-XA and LiRSI-Oxford datasets demonstrate the effectiveness and generalization ability of L2RSI. Future work will focus on reducing the reliance on additional magnetometers and further addressing the challenges of deploying this technology in real-world applications.

## 7 Acknowledgements

This work was partially supported by the Natural Science Foundation of China (No. 62471415), by the Fundamental Research Funds for the Central Universities (No. 20720230033), by Xiaomi Young Talents Program, by the Anhui Provincial Natural Science Foundation (No. 2508085MF142). We would like thank the anonymous reviewers for their valuable suggestions.

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

# A Appendix

## A.1 Limitations

While the proposed framework demonstrates strong performance in LiDAR place recognition over very large-scale remote sensing imagery and exhibits the potential to generalize across regions, we acknowledge two limitations of our work:

- Firstly, The current success of L2RSI requires external provision of a roughly accurate directional reference. Our collection device is equipped with a magnetometer that is calibrated with the LiDAR, providing a directional accuracy within $\pm 10$ degrees.

- Secondly, our approach is designed for urban scenes, and the performance may be affected in rural areas where the scene semantics are relatively sparse.

In future work, we will focus on overcoming this two limitations.

## A.2 Implementation details

The proposed L2RSI is implemented with Pytorch [49]. The network is trained with Adam optimizer [50] and experiences 100 training epochs. The learning rate is set to $5 \times 10^{-5}$, and the batch size is set to 64. We employ a StepLR learning rate scheduler, which reduces the learning rate by a factor of 0.95 every 1000 training steps to facilitate model convergence. All experiments are performed on Ubuntu 20.04.1 with a single NVIDIA GTX 3090 GPU with 24GB of memory.

## A.3 Comparison on more datasets

Table 8: Summary statistics for LiRSI-Kitti360.

| Submap Num. | Seq. 00 | Seq. 02 | Seq. 03 | Seq. 04 | Seq. 05 | Seq. 06 | Seq. 07 | Seq. 09 | Seq. 10 |
|---|---|---|---|---|---|---|---|---|---|
| PC | 2394 | 2090 | 241 | 1838 | 857 | 1456 | 791 | 1941 | 593 |
| RSI | 3132 | 5677 | 5677 | 5351 | 5351 | 5351 | 5233 | 7485 | 7485 |
| Area ($km^2$) | 2.93 | 6.30 | 6.30 | 6.98 | 6.98 | 6.98 | 8.58 | 6.06 | 6.06 |

KITTI-360 [7] is an urban scene dataset consisting of 9 sequences with a mileage of approximately $80km$, commonly used for 3D perception and localization. To enable a more comprehensive evaluation, we additionally associate each trajectory with corresponding remote sensing imagery. Following the same procedure described in Sec. 5.1, we expand it to LiRSI-Kitti360, with six sequences for training (02, 03, 04, 05, 06, 07), one sequence for validation (00), and two sequences for testing (09, 10). The statistical information of the dataset is summarized in Table 8.

Table 9: Performance comparison on LiRSI-Kitti360. $Recall@1$ and $Recall@10$ (%) are reported. The best result is rendered in bold.

| Methods | $Recall@1$ / $Recall@10$ (%) ↑ | | |
|---|---|---|---|
| | Seq. 00 | Seq. 09 | Seq. 10 |
| LIP-Loc [21] | 19.84/56.31 | 10.87/38.38 | 2.04/11.93 |
| GeoDTR [35] | 5.26/20.01 | 6.80/19.84 | 2.73/10.22 |
| Sample4Geo [36] | 43.98/80.16 | 33.08/65.84 | 14.82/34.58 |
| FRGeo [37] | 37.30/71.39 | 30.09/67.39 | 9.37/25.04 |
| L2RSI (w/o STPE) | 42.44/80.45 | 31.22/70.22 | 12.78/36.12 |
| L2RSI (w. SuperGlobal [48]) | 41.90/70.13 | 31.17/61.51 | 12.44/25.04 |
| L2RSI (w. PF) | 80.19/90.43 | 66.55/78.14 | 27.13/34.81 |
| L2RSI (w. STPE) | **91.09/96.12** | **92.28/98.15** | **42.38/56.13** |

The quantitative results are presented in Table 9. L2RSI (w. STPE) achieves the best performance across all sequences. Due to the differences in the coverage areas of the remote sensing databases, most methods perform slightly worse on Seq. 09 than in Seq. 00. The comparable performance of L2RSI (w. STPE) in Sequences 00 and 09 can be attributed to the use of $K = 30$ in the STPE algorithm, while L2RSI (w/o STPE) also has similar $Recall@30$ in both sequences. All methods exhibit noticeably poorer performance in Seq. 10, mainly cause the roadsides in this area are densely covered by trees in the remote sensing imagery, causing the roads to be heavily occluded by vegetation.

In contrast, the Velodyne HDL-64E LiDAR used in KITTI-360 has a vertical FOV of $+2°$ to $-24.8°$, which barely captures the vegetation above the road surface, leading to severe semantic inconsistency between the LiDAR and aerial imagery.

## A.4 Impact of different semantic encoders

Table 10: Ablation study about the selection of the semantic encoder. $Recall@1$ and $Recall@10$ (%) on LiRSI-XA and LiRSI-Oxford. † indicates that the input image size is $518 \times 518$. The best result is rendered in bold.

| Backbones | $Recall@1$ / $Recall@10$ (%) ↑ | | | | | |
| | Test Set-S | Test Set-M | Test Set-L | Test Set-G | 11-14-02-26 | 14-12-05-52 |
|---|---|---|---|---|---|---|
| VGG16 [51] | 73.77/88.04 | 62.83/73.27 | 51.39/69.96 | 47.45/62.83 | 21.60/38.86 | 20.82/36.85 |
| ResNet50 [52] | 59.38/76.99 | 51.26/67.01 | 48.93/61.60 | 43.52/55.20 | 13.56/23.27 | 11.73/22.82 |
| ResNet101 [52] | 72.58/87.70 | 68.02/81.80 | 62.49/76.02 | 54.62/66.18 | 24.66/41.26 | 26.87/43.21 |
| ConvNeXt-S [53] | 83.75/88.24 | 77.73/81.11 | 74.40/78.03 | 67.15/73.67 | 30.30/51.05 | 31.91/49.60 |
| DINOv2-S† [54] | 84.72/90.98 | 78.32/87.34 | 65.41/78.32 | 68.12/77.09 | 1.40/3.76 | 0.86/3.21 |
| MAE-B [47] | **88.93/92.13** | **87.95/91.64** | **85.49/90.65** | **83.27/87.82** | **42.29/59.41** | **43.77/62.76** |

We consider several popular backbones in the selection of the semantic encoder, including VGG [51], ResNet [52], ConvNeXt [53], DINOv2 [54] and MAE [47]. Table 10 shows the results evaluated on the test sets of LiRSI-XA and LiRSI-Oxford. Specific pre-trained model versions are selected based on their GPU memory requirements. MAE-B performs the best, especially in terms of generalization ability. We attribute this to mask modeling during training, which aids the model in interpreting the sparse semantic images generated from point cloud projections. DINOv2-S from the official source uses a default image size of $518 \times 518$. Images that are resized are marked with (†). Furthermore, to accommodate single-GPU training, the batch size is set to $8$, which hinder the model generalize to LiRSI-Oxford.

## A.5 Compatibility with existing pipelines

Table 11: Ablation study about the compatibility of STPE with other existing networks on LiRSI-XA and LiRSI-Oxford. $Recall@1$ and $Recall@10$ (%) are reported. The best result is rendered in bold.

| Methods | $Recall@1$ / $Recall@10$ (%) ↑ | | | | | |
| | Test Set-S | Test Set-M | Test Set-L | Test Set-G | 11-14-02-26 | 14-12-05-52 |
|---|---|---|---|---|---|---|
| Lip-Loc [21] | 52.89/77.86 | 46.86/69.74 | 42.07/60.64 | 32.23/46.74 | 4.07/18.69 | 3.58/18.68 |
| GeoDTR [35] | 47.52/67.81 | 39.40/55.26 | 32.41/45.06 | 11.33/17.85 | 1.97/5.46 | 1.11/5.67 |
| Sample4Geo [36] | **83.12/89.24** | **76.27/85.23** | **70.36/80.43** | **62.49/72.59** | **32.24/48.33** | **31.33/52.10** |
| FRGeo [37] | 81.34/85.55 | 68.94/77.80 | 62.32/72.28 | 53.95/63.06 | 7.95/16.87 | 7.98/17.39 |

The proposed STPE is easily compatible with existing place recognition pipelines. In Table 11, we refine the retrieval results of the baseline methods in using STPE. The results clearly indicate that all the methods achieve an improvement in recall on multiple test sets. However, when the vast majority of queries in the sequence fail completely, STPE is provided with insufficient useful information, as evidenced by the poor performance of Lip-Loc, GeoDTR and FRGeo on LiRSI-Oxford. Undoubtedly, when the relative position between LiDAR scans is available, the proposed STPE demonstrates superior effectiveness and compatibility with existing retrieval-based place recognition methods.

## A.6 Visualization

We present a visualization of the 100 square kilometer Test Set-G database in Fig. 6. As shown, over 60,000 remote sensing submaps are scattered across nearly all the major roads in the city. The proposed method does not require the assumption that the vehicle's trajectory is known, and it is capable of performing place recognition on all potential roads.

In Fig. 7, we present some qualitative comparisons of top-1 retrieval results on Test Set-G. Some challenging queries are selected. As can be seen, for most methods, even when the retrieval results are incorrect, they still tend to follow a road direction similar to that of the query. Especially in the fourth and sixth rows, where the query only shows roads and dense vegetation on both sides, the area within a $100km^2$ range contains numerous highly similar locations. Competitors are helpless, while the proposed L2RSI (w. STPE), by fully utilizing spatio-temporal information, achieves the correct Top-1 retrieval.

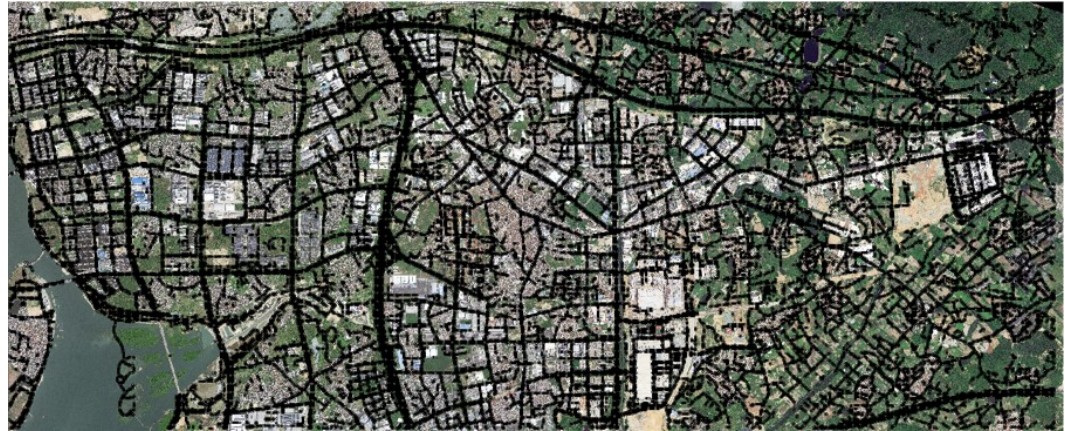

Figure 6: The visualization of the database on Test Set-G. The black circle marks the positions of the remote sensing submaps in the database.

The last row shows a failure case. Although our method has made a significant effort to find a highly similar location, including the east-west road, appropriate vegetation, buildings on both sides, and the road extending towards the upper right corner. However, in the reference remote sensing image, the south side of the correct location lacks buildings, which is due to the three-year temporal difference between the remote sensing image and the LiDAR data. In addition, we provide a video to more vividly showcase the place recognition results of L2RSI. The proposed method performs poorly on the east-west oriented road at the beginning of the video, which corresponds to the segment with serious temporal differences mentioned earlier.

## A.7   More details comparing with particle filtering

Table 12: The selection of particle number K for particle filtering. $Recall@1$ (%) on Test Set-S is reported. The runtime (ms) here only involves particle filtering. The best result is rendered in bold.

| Test set-S | 60 | 80 | 100 | 110 | 120 | 130 | 140 | 160 | 180 |
|---|---|---|---|---|---|---|---|---|---|
| $Recall@1$ (%) ↑ | 51.92 | 52.50 | 66.78 | 66.66 | **71.66** | 58.42 | 52.26 | 54.82 | 53.43 |
| $RT(ms)$ ↓ | 19.60 | **19.40** | 19.50 | 19.70 | 19.90 | 20.00 | 19.70 | 19.90 | 19.50 |

In the main paper, we mention the deployment of particle filtering in the retrieval framework to compare with the proposed particle estimation algorithm. Here we provide the specific details. We begin by generating the initial particles based on the top-K retrieval of the first query. When the second query arrives, we update the positions of particles using the relative pose between the two queries provided by Fast-GICP [38], along with the directional information from the magnetometer. Particles that fall within the $R = 30m$ of the top-K results of the second query are retained. If no particles survive, we regenerate new particles based on the top-K retrieval of the current query. Table 12 provides a performance comparison for different K. We ultimately chose $K = 120$ as the initial number of particles. In our tasks, due to unsatisfactory results for single queries, the particles need to be reset multiple times, which results in performance being weaker compared to the proposed STPE algorithm.

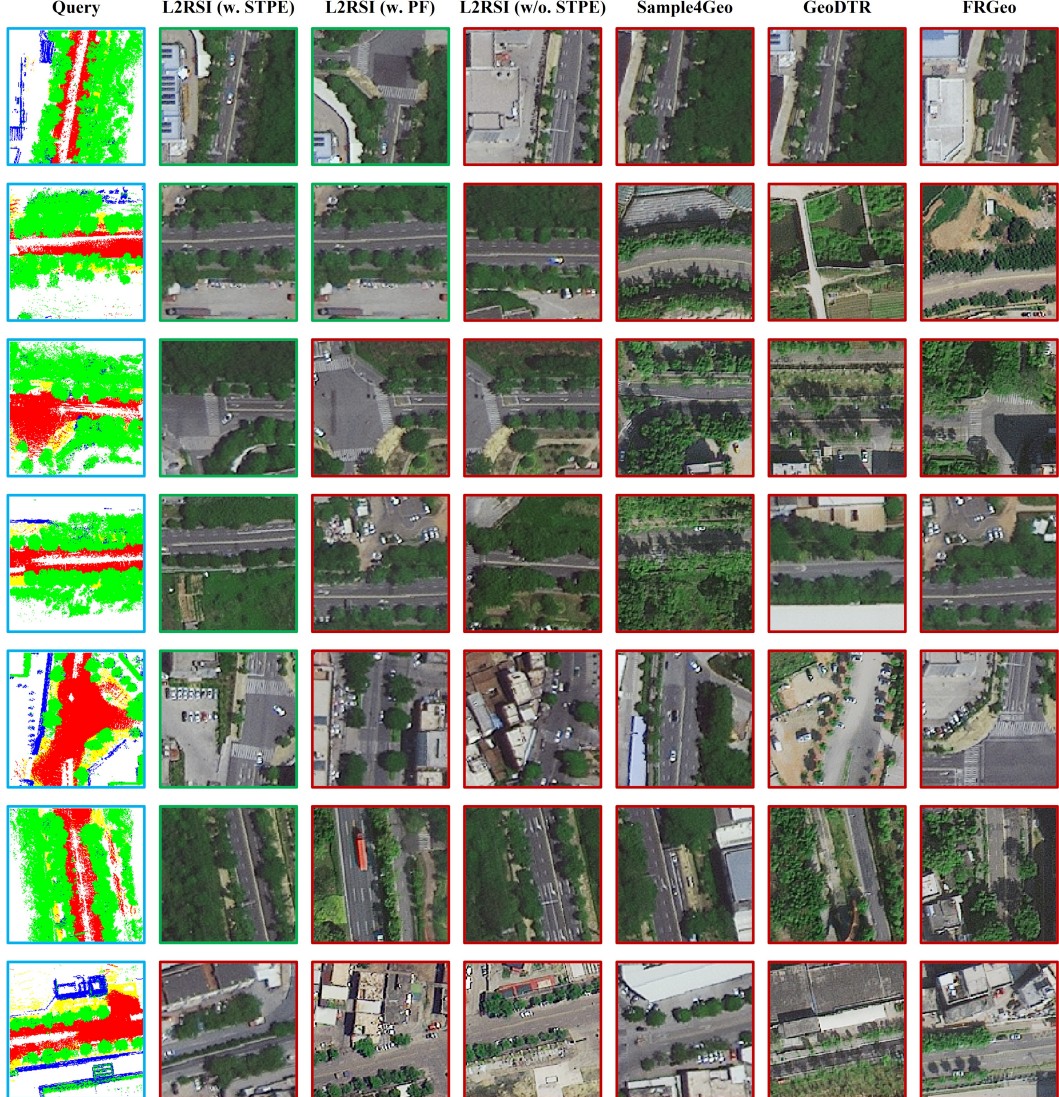

Figure 7: Qualitative comparisons on Test Set-G. Light blue boxes indicate the query point cloud submaps. Green boxes indicate correct retrieval results, while red boxes indicate incorrect results. To facilitate identification for the readers, we rotate the query to align with the same orientation as the remote sensing imagery.

