# OpenReview forum: "L2RSI: Cross-view LiDAR-based Place Recognition for Large-scale Urban Scenes via Remote Sensing Imagery"
_NeurIPS.cc/2025/Conference — NeurIPS 2025 poster_

### Official Review · Reviewer_4DTw · 2025-06-20

**Clarity:** 3
**Significance:** 2
**Originality:** 2
**Rating:** 4
**Confidence:** 4

**Summary:**

This paper addresses the problem of LiDAR-based localization in large-scale urban environments without relying on GPS or pre-built maps. It proposes a cross-view, cross-modality localization framework called L2RSI. The method achieves the matching between remote sensing imagery and LiDAR BEV (bird’s-eye view) maps through a framework that integrates semantic feature extraction, contrastive learning, and Spatial-Temporal Particle Estimation.

**Questions:**

See the weaknesses

**Ethical Concerns:**

["NO or VERY MINOR ethics concerns only"]

**Final Justification:**

The authors addressed key concerns with empirical results and detailed ablations. The STPE module shows practical value despite limited novelty, and the dataset will be open-sourced, enhancing impact. Some concerns remain (e.g., motion model sensitivity), but overall the paper offers a solid and useful contribution. I recommend borderline accept.

**Limitations:**

See the weaknesses

**Quality:**

3

**Strengths And Weaknesses:**

**Strengths:**

1.	The paper is well-structured and well-written. The proposed localization framework, which combines semantic feature extraction, contrastive learning, and Spatial-Temporal Particle Estimation, is easy to understand, and the experimental design is relatively rigorous.
2.	The engineering value is significant. The LiDAR + magnetometer-based localization solution eliminates the dependence on GPS or pre-built maps, making it practically meaningful for real-world engineering applications.
3.	Experiments are comprehensive. Extensive evaluations are conducted on two datasets, comparing the proposed method against multiple state-of-the-art approaches as well as different ablation variants of the method itself. The experimental design is well-structured and reasonable.

**weaknesses:**

1.	The completeness of this article is good, but the method itself lacks novelty and might be more suitable for submission to conferences or journals focused on engineering applications.

2.	The framework of “semantic feature extraction + contrastive learning” is not novel; this framework has been ubiquitously applied in the Place Recognition field.

3.	In Eq. 3 of the paper, the authors do not explain the motion model Mt-1, nor do they discuss the impact of the motion model's failure. If the FastGICP algorithm is used, consideration should be given to the interference from dynamic objects (such as pedestrians, vehicles, etc.) on motion estimation. Pose estimation drift would degrade the accuracy of point cloud submap generation, consequently reducing localization precision.

4.	A major highlight of the paper is the construction of the XA-L&RSI dataset. The lack of high-quality datasets is a significant impediment to progress in this field. However, the authors state in the checklist that this dataset will not be open-sourced, which limits the impact of this work.

---

> ### Author Rebuttal · Authors · 2025-07-31
>
> # Q1 and Q2: Lack of novelty in the method and framework.
>
> Thanks for your question. The main contribution of this work relies on two aspects.
>
> First, we attempt to build an ability that can be located immediately upon entering an unknown area for the first time, without GPS, without the pre-built map. To achieve this, our work has two settings that are significantly more challenging compared to traditional place recognition works: (1) High-resolution remote sensing imagery, which is easy to acquire, is applied for the first time as reference maps for retrieval in conjunction with lidar point cloud queries. (2) We have expanded the retrieval range to $100 km^2$, rather than searching along trajectories. For this cross-view, cross-domain retrieval problem, under such a strict setting, most previous works fail to provide satisfied performance, and we achieve a retrieval rate higher than $80\%$ for the first time.
>
> Second, at the technical level, our proposed retrieval framework based on semantic domains is a principled and effective solution, not a naive combination of existing modules. We conducted comprehensive ablation and comparative experiments on each component of the framework, demonstrating that our method provides a more robust and innovative integration (**Table 6 - Table9**). The core contribution lies in the proposed Spatial-Temporal Particle Estimation (STPE) module, which fully leverages both spatial and temporal continuity in sequential queries. This leads to a substantial improvement in retrieval accuracy, boosting Recall@1 from $20.07\%$ to $83.27\%$.
>
> Based on this, we believe that the exploration and technique contribution in this work will bring beneficial inspiration to the community.
>
> **Table 6: Performance comparison on XA-L&RSI and Oxford-L&RSI. (Table 3 in the main paper)**
>
> _Recall@1_ and _Recall@10_ (%)↑ are reported. The best result is rendered in **bold**.
>
> | Methods| Test Set-S| Test Set-M| Test Set-L| Test Set-G| 11-14-02-26| 14-12-05-52|
> |:--|:--:|:--:|:--:|:--:|:--:|:--:|
> | GeoDTR| 8.68 / 25.27 | 6.25 / 19.36 | 4.74 / 14.95| 4.39 / 9.26  | 0.47 / 3.18  | 0.48 / 3.71 |
> | Sample4Geo| 28.63 / 53.92 | 25.39 / 49.75| 24.23 / 46.27 | 22.95 / 43.02 | 7.13 / 20.68  | 7.51 / 21.75 |
> | FRGeo| 22.26 / 49.88 | 19.72 / 42.45 | 17.16 / 38.05 | 15.54 / 34.45     | 1.62 / 7.34  | 1.58 / 7.86 |
> | L2RSI (w/o STPE)| 30.05 / 68.56 | 23.90 / 61.95 | 21.93 / 57.42  | 20.07 / 52.55 | 10.19 / 30.23 | 11.25 / 30.58    |
> | L2RSI (w. SuperGlobal)  | 30.39 / 57.54| 23.90 / 53.02 | 22.27 / 48.26     | 20.19 / 42.92  | 8.90 / 22.98  | 8.98 / 22.86     |
> | L2RSI (w. PF) | 71.66 / 86.88| 55.17 / 72.47| 50.87 / 72.47 | 47.85 / 58.65 | 21.40 / 31.43 | 16.83 / 29.46    |
> | L2RSI (w. STPE)| **88.93 / 92.13** | **87.95 / 91.64** | **85.49 / 90.65** | **83.27 / 87.82** | **42.29 / 59.41**| **43.77 / 62.76**|
>
> **Table 7 : Ablation study about effectiveness of components. (Table 4 in the main paper)**
> Results include _Recall@1_ and _Recall@30_ (%)↑ on **Test Set-S** and **11-14-02-26** are reported.
> **Sem.**: Semantic segmentation. **STPE**: Spatial-Temporal Particle Estimation. **Orient.**: Orientation.
> The best results are in **bold**.
>
> | Sem. | STPE | Orient. | Test Set-S | 11-14-02-26|
> |:--:|:--:|:--:|:--:|:--:|
> || ✔ | ✔| 50.43 / 67.90  | 8.13 / 25.42  |
> | ✔    || ✔       | 30.05 / 85.50  | 10.19 / 45.90 |
> | ✔    | ✔    || 54.37 / 76.14| 4.92 / 21.24|
> | ✔| ✔ | ✔ | **88.93 / 95.94** | **42.29 / 70.15**|
>
> **Table 8: Ablation study about the selection of the semantic encoder. (Table 1 in Appendix)**
> _Recall@1_ and _Recall@10_ (%)↑ on XA-L&RSI and Oxford-L&RSI are reported.
> † indicates that the input image size is 518×518. The best result is rendered in **bold**.
>
> | Backbones | Test Set-S| Test Set-M | Test Set-L| Test Set-G| 11-14-02-26|14-12-05-52|
> |:--|:--|:--|:--|:--|:--|:--|
> | VGG16 | 73.77 / 88.04 | 62.83 / 73.27      | 51.39 / 69.96      | 47.45 / 62.83      | 21.60 / 38.86      | 20.82 / 36.85   |
> | ResNet50   | 59.38 / 76.99  | 51.26 / 67.01  | 48.93 / 61.60  | 43.52 / 55.20   | 13.56 / 23.27   | 11.73 / 22.82 |
> | ResNet101 | 72.58 / 87.70   | 68.02 / 81.80   | 62.49 / 76.02  | 54.62 / 66.18 | 24.66 / 41.26  | 26.87 / 43.21  |
> | ConvNeXt-S | 83.75 / 88.24  | 77.73 / 81.11  | 74.40 / 78.03  | 67.15 / 73.67  | 30.30 / 51.05   | 31.91 / 49.60   |
> | DINOv2-S† | 84.72 / 90.98   | 78.32 / 87.34  | 65.41 / 78.32  | 68.12 / 77.09  | 1.40 / 3.76    | 0.86 / 3.21   |
> | MAE-B | **88.93 / 92.13**  | **87.95 / 91.64**  | **85.49 / 90.65**  | **83.27 / 87.82**  | **42.29 / 59.41**  | **43.77 / 62.76**  |
>
> **Table 9: Ablation study about the compatibility of STPE with other existing networks. (Table 2 in Appendix)**
> _Recall@1_ and _Recall@10_ (%)↑ on XA-L&RSI and Oxford-L&RSI are reported. The best result is rendered in **bold**.
>
> | Methods | Test Set-S| Test Set-M| Test Set-L | Test Set-G| 11-14-02-26| 14-12-05-52|
> |:--|:--:|:--|:--|:--|:--|:--|
> | GeoDTR  | 47.52 / 67.81      | 39.40 / 55.26      | 32.41 / 45.06      | 11.33 / 17.85      | 1.97 / 5.46        | 1.11 / 5.67        |
> | Sample4Geo   | **83.12 / 89.24**  | **76.27 / 85.23**  | **70.36 / 80.43**  | **62.49 / 72.59**  | **32.24 / 48.33**  | **31.33 / 52.10**  |
> | FRGeo                 | 81.34 / 85.55      | 68.94 / 77.80      | 62.32 / 72.28      | 53.95 / 63.06      | 7.95 / 16.87       | 7.98 / 17.39       |
>
> # Q3: The impact of the motion model's failure.
>
> As noted in Lines 205–209 of the main paper, the motion model for each time step} is directly obtained using FastGICP combined with magnetometer-based heading correction.
>
> We acknowledge that dynamic objects (e.g., vehicles, pedestrians) can indeed introduce noise to motion estimation. To mitigate this, we deliberately avoided prolonged proximity to large moving vehicles during data collection. Additionally, we filtered out point clouds within a 3-meter radius of the data collection vehicle to reduce self-interference.
>
> Empirically, the submaps constructed via FastGICP show high quality in visualizations, and the particle aggregation mechanism in STPE over multiple time steps helps to neutralize occasional noisy estimates. This robustness is also reflected in the strong performance of place recognition observed in our experimental results (**Table 6** and **Table 7**).
>
> # Q4: The XA-L\&RSI dataset open source.
>
> Thanks for acknowledging the value of our dataset. We fully agree with your view on the importance of open data, especially given the scarcity of datasets in this research area — a challenge we are also deeply aware of. Before submitting the paper, we only acquired a usage license (rather than an open license) for the high-resolution ($0.5m$) satellite imagery of Xiang'an District, Xiamen, which prevented us from releasing the dataset in the short term. This was clearly stated in the checklist.
>
> However, we are pleased to share that the licensing issues have recently been resolved. We now hold the appropriate rights to release the imagery, and we plan to open-source the XA-L\&RSI dataset upon acceptance of the paper.

---

> ### Author Response · Authors · 2025-08-05
>
> Dear Reviewer 4DTw,
>
> Thanks for your effort in reviewing our work. We have addressed your concerns in the rebuttal and summarize our response as follows:
>
> 1. Lack of novelty in the method and framework (**W1, W2**): We have clarified that the primary contribution of our work lies in both the problem setting and the proposed framework. **we are the first to address the challenge of large-scale (over 100 km²) urban cross-view LiDAR-based place recognition with high-resolution remote sensing imagery.** The integration of Spatial-Temporal Particle Estimation (STPE) provides a principled and innovative solution by exploiting both spatial and temporal continuity in sequential queries, which significantly improves retrieval accuracy (**Recall@1 increased from 20.07% to 83.27%**).
>
> 2. The impact of the motion model's failure (**W3** ):
> We explained the motion model estimation method, discussed how we mitigate the interference of dynamic objects on the FastGICP algorithm in practice, and the experimental results in the main paper also demonstrate the robustness of the STPE algorithm to random noise in the motion model.
>
> 3. The XA-L&RSI dataset open source (**W4** ):
> We have clarified the reason for filling in "Data cannot be open sourced in the short term." in the checklist, and we also confirmed that the datasets used in our work will be made publicly available to the research community upon acceptance of the paper.
>
> ### **Additional Experiment on Motion Model's Failure(W3)**:
> We have just completed an additional experiment to address your concerns about **the impact of the motion model's failure**. We introduce different levels of **yaw angle noise $\epsilon_{yaw}(^\circ)$** and **translation noise $\epsilon_{xy}(m)$** into the estimated motion model. The translation noise is defined by the noise in the x-direction and y-direction as follows:
> $$\epsilon_{xy}=\sqrt(\epsilon_x^2+\epsilon_y^2)$$
>
> In this experiment, we take $\epsilon_x=\epsilon_y$ for convenience. The average Recall@1(%) of three repeated experiments on Test Set-G ($100km^2$) of XA-L&RSI are shown in the following table:
>
> |**Recall@1(%)**|**$\epsilon_{yaw}=\pm0^\circ$**|**$\epsilon_{yaw}=\pm5^\circ$**|**$\epsilon_{yaw}=\pm10^\circ$**|**$\epsilon_{yaw}=\pm15^\circ$**|**$\epsilon_{yaw}=\pm30^\circ$**|
> |-|-|-|-|-|-|
> |**$\epsilon_{xy}=\pm0$m**|83.27(-0.00)|80.57(-2.70)|78.72(-4.55)|60.76(-22.51)|48.46(-34.81)|
> |**$\epsilon_{xy}=\pm1$m**|82.77(-0.50)|80.20(-3.07)|77.85(-5.42)|59.77(-23.50)|44.26(-39.01)|
> |**$\epsilon_{xy}=\pm4$m**|71.58(-11.69)|67.04(-16.23)|66.91(-16.36)|56.70(-26.57)|41.39(-41.88)|
> |**$\epsilon_{xy}=\pm9$m**|67.04(-16.23)|64.94(-18.33)|61.50(-21.77)|54.37(-28.90)|39.12(-44.51)|
> |**$\epsilon_{xy}=\pm16$m**|57.31(-25.96)|53.87(-29.40)|53.75(-29.52)|50.92(-32.35)|35.91(-47.36)|
>
> **Please note that the disturbances in the table represent intentionally introduced additional noise, based on the noise inherently caused by the FastGICP algorithm and the magnetometer used in the system.**
>
> $\epsilon_{yaw}=\pm5^\circ$ and $\epsilon_{xy}=\pm1$m indicates that, in addition to the inherent noise, random yaw angle noise within a range of $\pm5^\circ$ and random translation noise within a range of $\pm1m$ are introduced.
>
> **The parentheses report the performance drop compared to the additional noise-free setting.**
>
> The results demonstrate that our method can tolerate a certain level of additional noise in the estimated motion model ($\epsilon_{yaw}<\pm10^\circ$ and $\epsilon_{xy}<\pm1$m), which is typically introduced by dynamic objects affecting the FastGICP algorithm. **In practice, we found that excluding the points of the data collection vehicle itself from the raw point cloud effectively reduces this noise to an acceptable level.**
>
> **We are eager to engage in further discussion if you have any additional concerns.**
>
> Best Regards,
>
> Authors

---

> > ### Comment · Reviewer_4DTw · 2025-08-06
> >
> > The authors have provided a thorough response with compelling empirical results and extensive ablations.
> >
> > While the core framework builds on established paradigms, the integration of semantic priors with the proposed STPE module demonstrates solid engineering insight and delivers clear performance gains under challenging settings.
> >
> > The resolution of the dataset licensing issue also strengthens the paper's contribution. I am therefore raising my score.

---

> > > ### Author Response · Authors · 2025-08-06
> > >
> > > Thank you for raising your score and for your encouraging comment. We sincerely appreciate your recognition of our efforts in addressing your concerns.
> > >
> > > In the camera-ready version:
> > >
> > > 1. We will include the additional experiments on Motion Model's Failure in the main paper and provide a detailed discussion of its impact on the retrieval process.
> > >
> > > 2. We will also provide the dataset release link in Section 5.1 (Benchmark Datasets).
> > >
> > > We are grateful for your constructive feedback throughout the review process, which has been instrumental in enhancing the quality and impact of our work.

---

### Official Review · Reviewer_HoHD · 2025-06-28

**Clarity:** 2
**Significance:** 2
**Originality:** 2
**Rating:** 5
**Confidence:** 3

**Summary:**

The paper addresses place recognition between a lidar point cloud query and a database of remote sensing image (i.e. Bird Eye View (BEV) aerial sattelite or drone images).

The contribution is a network that generates embeddings in a shared space for both the point cloud and the BEV images.
The network can take as input either the BEV aerial image or a BEV rasterization of the point cloud.
To make the embedding even more informative, the network is fed a semantic segmentation of either input: a standard semantic of the BEV aerial image; the point cloud is first segmented then rasterized.

The network is trained with the standard InfoNCE loss.

To demonstrate the relevance of the method, the paper further contribute a training and testing dataset made of aerial images and point cloud submaps.
Part of the dataset is used for training their method and another non overlapping part is used for testing, that covers 100km2, which emulates a large-scale place recognition scenario.

The paper further completes the existing Oxford retrieval dataset with aerial images, as that already provides point cloud data but not aerial data.
This dataset serves to demonstrate the generalizability of the trained network.

The paper is compared against cross-view image-based place recognition i.e., natural images against a database of aerial images.
The proposed method outperforms the baselines demonstrating the relevance of the proposed method.

**Questions:**

- What augmentations are applied to the images? Are they rotated along the vertical axis to augment their robustness to orientation variations?

**Ethical Concerns:**

["NO or VERY MINOR ethics concerns only"]

**Final Justification:**

Thank you for the explanations, they are very useful and clarify how the estimator works.

These details are important and deserve to be in the main paper (some space may be saved by writing some of the equations in lines with the text).

Similarly, reviewer 4DTw raised an important problem regarding the reliance of the "retrieval temporal estimation" on the motion model and how it can be "impacted by the motion model's failure". The additional experiments in the authors' answer that simulate such failure are very relevant and worth adding in the paper (or the supplementary).

The rebuttal addresses the main limitations that motivated the previous decision:

related work: some of the references were missed in the review, I apologize
comparison with other lidar to aerial image retrieval: it appears that previous such methods did not release their code so a comparison is not possible without an important engineering effort, so it is acceptable that these comparisons are missing
temporal estimation of the retrieval result: the derivation is better explained in the rebuttal and these details should be included in the paper.
It would be useful to explicitly mention how the proposed temporal estimation differs from previous temporal or sequence-based methods for example:

Dynamically Modulating Visual Place Recognition Sequence Length For Minimum Acceptable Performance Scenarios C Malone, A Vora, T Peynot, M Milford, IROS 2024
Seqmatchnet: Contrastive learning with sequence matching for place recognition & relocalization S Garg, M Vankadari, M Milford - Conference on Robot Learning, 2022
and several others
Since the rebuttal addresses the main limitations and that the dataset contributed in this paper and that the temporal estimation used in the retrieval is sound and useful (see Tab3 L2RSI vs L2RSI with STPE), the rating is raised to accept.

**Limitations:**

No, the authors have not adequately addressed the limitations of their work.
There is no potential negative societal impact of their work

Evaluation:
- The threshold used in the recall definition is 30 meters which seems to small given the resolution of the input images (that cover 60x60 meters). What is the query point cloud is 40 meter away from the center of the closest submap? Then it is within a submap that should be retrieved but the recall will still be 0.

Misc comments:
- L192: "top-C" should be "top-K"?

**Quality:**

2

**Strengths And Weaknesses:**

S1. The problem of cross-modality (lidar vs image) and cross-view (natural lidar cloud vs aerial images) place recognition is relevant to the research community and the industry (autonomous systems and driving).

S2. The method is simple and efficient, making it reasonnable to deploy.

S3. The paper provides annotated data (pairs of lidar clouds and their associated aerial images) for training and testing.


W1. The related work is incomplete. Here is a non exhaustive list of missing references:

[1] Tang, Tim Y., Daniele De Martini, and Paul Newman. "Get to the point: Learning lidar place recognition and metric localisation using overhead imagery." Proceedings of Robotics: Science and Systems, 2021 (2021).

[2] Shubodh, Sai, et al. "Lip-loc: Lidar image pretraining for cross-modal localization." Proceedings of the IEEE/CVF Winter Conference on Applications of Computer Vision. 2024.

[3] Xu, Huaiyuan, et al. "C2L-PR: Cross-modal camera-to-LiDAR place recognition via modality alignment and orientation voting." IEEE Transactions on Intelligent Vehicles (2024).

[4] Lee, Alex Junho, et al. "$^{2} $: LiDAR-Camera Loop Constraints for Cross-Modal Place Recognition." IEEE Robotics and Automation Letters 8.6 (2023): 3589-3596.

[5] Komorowski, Jacek, Monika Wysoczańska, and Tomasz Trzcinski. "Minkloc++: lidar and monocular image fusion for place recognition." 2021 International Joint Conference on Neural Networks (IJCNN). IEEE, 2021.

[6] Li, Yun-Jin, et al. "VXP: Voxel-Cross-Pixel Large-Scale Camera-LiDAR Place Recognition." International Conference on 3D Vision 2025.

[7] Guan, Tianrui, et al. "Crossloc3d: Aerial-ground cross-source 3d place recognition." Proceedings of the IEEE/CVF International Conference on Computer Vision. 2023.

[8] Fervers, Florian, et al. "Continuous self-localization on aerial images using visual and lidar sensors." 2022 IEEE/RSJ International Conference on Intelligent Robots and Systems (IROS). IEEE, 2022.

[9] Tang, Tim Yuqing, Daniele De Martini, and Paul Newman. "Point-based metric and topological localisation between lidar and overhead imagery." Autonomous Robots 47.5 (2023): 595-615.

W2. The evaluation compares only against cross-view (natural images vs aerial images) place recognition but not against cross modality (lidar vs aerial images) methods, which makes the evaluation severely incomplete.
Relevant methods for which code is available include [2], [3], [4]


W3. The mathematical derivation in the particle-filter-based filtering of the retrieval candidate are confusing:
- the state of the particle is not defined (the reader can guess that it contains the x,y coordinate of the system only?)
- L201: What does "aggregating the particles" mean? That a new set of particle is added at time t to the previous obtained at t-1 by applying the motion model to all particles of t-1 and saving the particle before and after the motion?
- Eq4 is incorrect: the conditioning variable $x_{t-1}$ replacement is not explained
- There is no link between equations (4) and (5)
- The Gaussian distributions starting Eq6 all miss their covariances
- the motion step of the filter in eq10 implies that there is no noise since only the means are updated but not the covariance whereas Eq3 defines a noise v in the moition model


Given that important baselines are missing from the evaluation (there is not comparison against lidar-vs-image place recognition methods, the baselines are all image-vs-image methods that were retrained on the lidar-vs-image dataset), the incomplete related work and the confusion in the proposed filtering method, the paper does not pass the acceptance bar.

---

> ### Author Rebuttal · Authors · 2025-07-30
>
> # Q1: The related work is incomplete.
> Thanks for raising this concern. Firstly, among the nine references mentioned as missing, four are already cited in the introduction or related work sections, including (LC)$^2$ [4], VXP [6], Tang et al. [9], and the journal version of Tang et al. [1].
>
> Secondly, the motivation of this work is to develop a capability for immediate localization upon first entry into an unknown environment, without relying on GPS or a pre-built 3D map. Therefore, we choose globally available and easily accessible remote sensing imagery as the reference. In terms of related work, we focus on cross-view and cross-modal retrieval. The remaining five papers in the list are as follows:
>
> As shown in **Table 3**, since [2], [3], [5], and [7] all rely on a pre-built 3D map of the target area constructed from ground-level or low-altitude observations, they are not directly related to our work.
>
> [8] is more accurately described as a geo-tracking method. It requires an initial absolute pose and uses multiple ground cameras and a LiDAR sensor to match against a small ($100\times100m^2$) satellite image. This setup is completely unrelated to our problem.
>
> **Table 3: The task settings of the mentioned Related Works.**
>
> |Methods|Query|Reference|
> |---|---|---|
> | [2], [3]  | Ground Camera   | Ground LiDAR |
> | [5]| Ground Camera + Ground LiDAR | Ground Camera + Ground LiDAR |
> | [7] | Ground LiDAR | Aerial LiDAR |
> | Ours | Ground LiDAR | Remote sensing image  |
>
> In summary, since these works are not strongly related to our approach, we did not cover the entire set of literature from those fields. Nevertheless, we will include [2], [3], [5], and [7] in the Related Work of the camera-ready version.
> # Q2: Lack of comparison with the methods of Lidar to aerial images.
> **Table 4: Supplementary performance comparison.**
>
> $Recall@1$ and $Recall@10$ (%)↑ are reported.
> | Methods |Test Set-S | Test Set-M| Test Set-L| Test Set-G  | 11-14-02-26| 14-12-05-52|
> |---|---|---|---|---|---|---|
> |PointLoc [1]| 10.37/27.63 | 7.89/24.61| 6.91/22.32| 6.52/19.85| 0.92/7.63| 1.01/8.14|
> | LIP-Loc (official) [2] | 3.13/10.32 | 1.74/8.12| 1.51/7.19| 0.46/3.94 | 1.18/5.48 | 1.26/6.10|
> | LIP-Loc (BEV) [2]  | 16.82/53.48 | 13.69/45.36| 11.60/41.65 | 11.02/39.44| 0.88/10.78| 0.96/11.13  |
> | L2RSI (w/o STPE) | 30.05/68.56 | 23.90/61.95 | 21.93/57.42  | 20.07/52.55 | 10.19/30.23 | 11.25/30.58 |
> | L2RSI (w. STPE)  | **88.93/92.13** | **87.95/91.64** | **85.49/90.65**| **83.27/87.82** | **42.29/59.41** | **43.77/62.76** |
>
> Thanks for the suggestion. However, the recommended baselines [2], [3], and [4] address the place recognition between LiDAR and street-view camera rather than LiDAR and aerial image, and are not directly related to our work.
>
> Among them, [3] and [4], the proposed methods are tightly coupled with monocular depth estimation from street-view imagery. However, high-quality depth estimation is not feasible for satellite imagery due to the lack of reliable geometric cues. As a result, these methods cannot be directly applied to our setting, and a fair comparison is not possible.
>
> [2] projects LiDAR into a range image and then performs image-to-image retrieval against street-view imagery. Given the substantial viewpoint discrepancy between range images and satellite imagery, along with the constraint in [2] that restricts retrieval to the vehicle's trajectory, it is expected that directly transferring this method to large-scale, city-level retrieval would result in poor performance. In **Table 4**, we retrained and compared both the official implementation of [2] and a variant where the range image is replaced with a magnetometer-corrected semantic BEV image in XA-L\&RSI, to provide a more fair and meaningful comparison. They both significantly underperform our method across all benchmarks.
>
> It is important to highlight that, to the best of our knowledge, our work is the first to tackle LiDAR-to-Remote-Sensing-Imagery place recognition in a city-scale area ($100km^2$). In the list of **Q1**, [1] and [9] are the most relevant. Unfortunately, neither of them is open-sourced, and [9] does not even provide a network architecture diagram, which significantly increases the difficulty of reproduction.
>
> Moreover, both [1] and [9] are designed for trajectory-constrained retrieval, meaning they assume localization occurs only along pre-defined driving routes. As a result, their performance degrades quickly as trajectory length increases, which limits their applicability in large-scale, unconstrained scenarios.
>
> During the rebuttal period, we made our best effort to re-implement the approach proposed in [1], and the results are reported in **Table 4**. The performance remains substantially below that of our method.
> # Q3: Mathematical derivation of STPE.
> Thank you for carefully reviewing our mathematical derivation.
>
> - **Yes**, the state $\textbf{x}_j^{[k]}$ of a particle refers to the 2D coordinate $(x, y)$ of a candidate submap in the remote sensing database retrieved by the query $Q_j$.
>
> - By "aggregating the particles", we refer to propagating particles from multiple past time steps *(t−L to t−1)* to time step *t* using the motion model, and then constructing a probability density function at time step *t* based on the combined particle distribution.
>
> - We agree that Eq. 4 and Eq. 5 may cause confusion. Our original intention was to express, in probabilistic terms, how the particles from time step *t−1* transition to time step *t*. However, as described from L205 to L209 of the main paper, the motion of the particles are directly obtained using FastGICP and magnetometer-based heading correction. Therefore, we have removed Eq. 4 and Eq. 5 to avoid ambiguity. This change does not affect the overall flow of the derivation.
>
> - We assume the x and y dimensions are independent, and thus the diagonal covariance matrices are used in Eq.6 and subsequent Gaussian mixture models :
> $$
> \Sigma =
> \left[
> \begin{array}{cc}
> \sigma_x^2 & 0 \\\\
> 0 & \sigma_y^2
> \end{array}
> \right]
> $$
>
> - The noise term v in Eq. 3 originates from the motion estimation process based on FastGICP combined with magnetometer-based heading correction. This procedure inherently produces noisy estimates of the motion model. When aggregating many particles over multiple time steps, the effect of such noise tends to average out. Therefore, we model the distribution at time t empirically as a mixture of multiple gaussian models without explicitly including a separate noise term in Eq. 10. Empirically, this approximation works well in our experiments. (Table 3 and Table 4 in the main paper, **please see Table 6 and Table 7 included in the response to Reviewer 4DTw for convenient reference.**)
>
> # Q4: Data enhancement.
> Yes, we applied data augmentation. Specifically, we performed random rotations along the z-axis within $\pm 15^\circ$ to simulate heading noise and improve robustness to magnetic disturbances.
> # Q5: Limitations.
> Thank you for the comment. As noted in the Limitations (Appendix A.1), our method relies on magnetometer-based heading priors and is designed for urban environments. Performance may degrade in semantically sparse rural areas. We acknowledge these limitations and plan to address them in future work.
> # Q6: Threshold of $30m$ in evaluation.
> **Table 5: Ablation study about the distance threshold of Retrieval θ.**
>
> $Recall@1$ and $Recall@10$ (%)↑ are reported.
> | θ| Methods| Test Set-S|Test Set-M| Test Set-L| Test Set-G| 11-14-02-26|14-12-05-52|
> |:--|:--|:--|:--|:--|:--|:--|:--|
> |30| L2RSI (w/o STPE) | 30.05 / 68.56| 23.90 / 61.95| 21.93 / 57.42| 20.07 / 52.55 |10.19 / 30.23| 11.25 / 30.58|
> || L2RSI (w. STPE)  | 88.93 / 92.13| 87.95 / 91.64| 85.49 / 90.65| 83.27 / 87.82|42.29 / 59.41| 43.77 / 62.76 |
> |40| L2RSI (w/o STPE) | 36.54 / 73.78| 29.81 / 70.77| 27.38 / 66.01| 25.06 / 62.41| 45.33 / 63.71| 45.31 / 64.98|
> || L2RSI (w. STPE)  | **90.04 / 97.17**| **88.19 / 95.08**| **87.82 / 94.46**| **86.47 / 91.64**| **12.73 / 33.18**| **13.88 / 34.29**|
>
> We appreciate the reviewer’s suggestion. The choice of a distance threshold $\theta=30m$ is a deliberate trade-off. During training, a larger threshold reduces the spatial overlap between LiDAR and positive remote sensing images, which weakens the effectiveness of contrastive learning. A smaller threshold, however, requires denser remote sensing submap sampling to ensure every LiDAR query has valid positives.
>
> During inference, increasing the threshold simplifies the retrieval task and leads to higher recall, but with weaker spatial alignment. Our $30m$ threshold ensures at least $50\%$ overlap between LiDAR BEV images and positive remote sensing images to reflect a more realistic and challenging setting.
>
> For reference, we provide results under a threshold $\theta=40m$ in **Table 5**, which show higher recall due to relaxed retrival criteria.
>
> # Q7: L192: top-C should be top-K?
> Thank you for the question. We intentionally use "top-C" to refer to the number of retrieval candidates produced by global place recognition for a single current query ($C=200$ in our setting). In contrast, "K" denotes the number of top retrieval results selected from each historical query in the sequence during the STPE refinement stage ($K=30$ in our setting). If $L$ denotes the number of historical queries in the sequence, the STPE module operates on $K\times L$ particles in total and re-ranks the top-C retrieval candidates of the current query.

---

> > ### Comment · Reviewer_HoHD · 2025-08-03
> > **Relevant rebuttal but misses clarification on the particle filter still**
> >
> > Thank you for the rebuttal that clarifies:
> > - the link between the paper and related work (this paper is not the first to address lidar to aerial image place recognition but the previous methods did not release their code)
> > - the derivation of the particle filter. However, it would be good to have a clear and explicit description of the math derivation since it is one of the main contribution of the paper (together with the dataset): the retrieval performance of the network alone are much lower than with the "sequence-based" retrieval enabled by the particle filter estimation.
> >
> > A few other questions remain on the motion model used in the particle filter estimation. From the paper, it seems that when running the retrieval on query $Q_j^t$, the previous $L$ past queries $Q_j^{t2}$ with ${t2} \in {t-1, t-L}$, are also used. The retrieval process has already run on those past queries to retrieve the top-C database images. These database images are then used to define a probability distribution of the position of $Q_j^t$.
> > - how is the retrieval on the first query run i.e. the initialization step? at t=0, the range [t-1, t-L] is not valid
> > - why are the retrieval candidates for $Q_j^t$ defined as the aerial images with the most similar network embedding vector not included in the derivation?
> > - all the database images of the past  $L$ past queries $Q_j^{t2}$ with ${t2} \in {t-1, t-L}$ define the particles used in the particle estimation. How are they initialized? For example, are the particle used at time t initialized with their last value during the estimation at t-1? Or are they initialized with the position associated to the retrived aerial image?

---

> ### Author Response · Authors · 2025-08-04
> **Clarifications and Responses to the Comments**
>
> Thank you for your thoughtful and detailed feedback on our rebuttal.
> ___
> ## Main Comments:
> > **the link between the paper and related work (this paper is not the first to address lidar to aerial image place recognition but the previous methods did not release their code)**
>
> Thank you for recognizing our efforts during the rebuttal period. We do not claim ours is the first one to address lidar to aerial image place recognition. But we want to emphasize that **we are the first to address the challenge of large-scale (over 100 km²) urban cross-view LiDAR-based place recognition with high-resolution remote sensing imagery.** The previous methods [1] and [9] differ from ours in a fundamental way: **They limit the retrieval range to a predefined trajectory instead of an open geographic area**, assuming a known driving route, making these methods impractical for real-world use.
>
> > **the derivation of the particle filter....**
>
> We would like to clarify the difference between the STPE algorithm and the traditional particle filter. The particle filter relies on an initial particle set that decays over time steps, requiring reinitialization once the particles disappear.
>
> In contrast, the STPE algorithm operates differently. In our approach, particles are maintained in a queue. For each current query, only the retrieval results of the previous $L$ queries, including the current query itself, are used as particles for propagation. This mechanism ensures the smooth transmission of spatio-temporal information, without the need for reinitialization.
>
> > **Q1: How is the retrieval on the first query run i.e. the initialization step? at t=0, the range [t-1, t-L] is not valid**
>
> **A1:** Thanks for your insightful question. The STPE algorithm is not applicable at $t=0$. In fact, the results reported in the paper all begin from $t=L-1$, which does not impact fairness. In practical applications, the target area is an unknown region that has never been visited before, and our ultimate goal is to achieve accurate localization right from the start. However, currently, a small distance (around 250m) must be traveled first, during which only single-query retrieval results can be provided.
>
> > **Q2: Why are the retrieval candidates for $Q_j^t$ defined as the aerial images with the most similar network embedding vector not included in the derivation?**
>
> **A2:** Thanks for your careful review. In the main paper, we stated that the time steps in the STPE algorithm sequence range from $t-L$ to $t-1$. However, the correct range should be from $t-L+1$ to $t$. The total number of queries in the sequence remains $L$. We sincerely apologize for the misunderstanding caused by the typographical error. We will correct the derivation and description in Section 4.3 in the camera-ready version, including a revision of Eq. 11 as follows:
>
> $$P_t(x, y) = \frac{1}{L} \sum_{j=t-L+1}^t \tilde{P}_j(x, y)$$
>
> > **Q3: all the database images of the past past L queries $Q_j^{t2}$ with $t2 \in t-1, t-L$ define the particles used in the particle estimation. How are they initialized? For example, are the particle used at time t initialized with their last value during the estimation at t-1? Or are they initialized with the position associated to the retrived aerial image?**
>
> **A3:** Thank you for your insightful question. The former is mostly correct: the particles used at time $t$ are initialized with their values from the estimation at time $t-1$. However, since our particle set is maintained as a queue, the particles at time $t$ are updated by adding the retrieval candidates of current query itself, while removing the particles from time $t-L$.
> ___
> Again, thank you for taking the time and energy to help us improve the paper.

---

> > ### Comment · Reviewer_HoHD · 2025-08-05
> > **The temporal estimation of the retrieval is clearer and should be added to the paper**
> >
> > Thank you for the explanations, they are very useful and clarify how the estimator works.
> >
> > These details are important and deserve to be in the main paper (some space may be saved by writing some of the equations in lines with the text).
> >
> > Similarly, reviewer 4DTw raised an important problem regarding the reliance of the "retrieval temporal estimation" on the motion model and how it can be "impacted by the motion model's failure".
> > The additional experiments in the authors' answer that simulate such failure are very relevant and worth adding in the paper (or the supplementary).
> >
> > The rebuttal addresses the main limitations that motivated the previous decision:
> > - related work: some of the references were missed in the review, I apologize
> > - comparison with other lidar to aerial image retrieval: it appears that previous such methods did not release their code so a comparison is not possible without an important engineering effort, so it is acceptable that these comparisons are missing
> > - temporal estimation of the retrieval result: the derivation is better explained in the rebuttal and these details should be included in the paper.
> >
> > It would be useful to explicitly mention how the proposed temporal estimation differs from previous temporal or sequence-based methods for example:
> > - Dynamically Modulating Visual Place Recognition Sequence Length For Minimum Acceptable Performance Scenarios
> > C Malone, A Vora, T Peynot, M Milford, IROS 2024
> > - Seqmatchnet: Contrastive learning with sequence matching for place recognition & relocalization
> > S Garg, M Vankadari, M Milford - Conference on Robot Learning, 2022
> > - and several others
> >
> > Since the rebuttal addresses the main limitations and that the dataset contributed in this paper and that the temporal estimation used in the retrieval is sound and useful (see Tab3 L2RSI vs L2RSI with STPE), the rating is raised to accept.

---

> > > ### Author Response · Authors · 2025-08-06
> > >
> > > Thank you for raising your score and for your encouraging feedback. We truly appreciate you recognizing the improvements we have made.
> > >
> > > We will thoughtfully incorporate your suggestions and make the necessary revisions to both the main paper and the supplementary material in the camera-ready version to further enhance the clarity and overall quality of the work.
> > >
> > > Specifically,
> > >
> > > 1) We will focus on improving the mathematical notation, derivations, and corresponding explanations in Section 4.3, and incorporate the revisions mentioned during the rebuttal into the main paper.
> > >
> > > 2) We will include the simulation experiments on motion model failure in the main paper, along with additional descriptions to better explain the motion model and its impact on the retrieval process.
> > >
> > >
> > > 3) Regarding the two prior sequence-based methods you mentioned, they operate in a **ground camera sequence to ground camera sequence retrival,** which introduce sequence information in the training of the retrieval network. This is a good idea and provides meaningful inspiration.
> > >
> > >     In the current setting, sequence-based retrieval methods are difficult to be directly applied, **as the reference remote sensing imagery is inherently static and lacks temporal continuity or directional cues.** Applying such methods would require substantial engineering efforts to construct a sequence-like data structure from the remote sensing imagery.
> > >
> > >     However, we will include a discussion on these methods in the related work section to explicitly clarify the differences. Moreover, we consider that sequence-to-sequence retrieval is a promising future direction, and we plan to explore it in our subsequent work.
> > >
> > > Once again, thank you for the time and effort you have dedicated to helping improve our work.

---

### Official Review · Reviewer_z1Y6 · 2025-06-30

**Clarity:** 2
**Significance:** 3
**Originality:** 3
**Rating:** 5
**Confidence:** 3

**Summary:**

This work introduces a novel method for cross-view LiDAR-based place recognition. The proposed approach extracts semantic maps from aerial images and LiDAR BEV scans, leveraging a contrastive learning framework to align features in a shared semantic space. Additionally, the authors introduce a probability propagation method based on particle estimation to refine position predictions. Experimental results on two self-collected datasets demonstrate significant performance improvements over existing methods.

**Questions:**

See weakness. Section 3.4 could be improved for easy understanding.

**Ethical Concerns:**

["NO or VERY MINOR ethics concerns only"]

**Final Justification:**

The authors have addressed most of my concerns. Therefore, I will raise my score.

**Limitations:**

Limitations regarding data coverage, dependent segmentation models should be discussed.

**Paper Formatting Concerns:**

Table order can be adjusted.

**Quality:**

3

**Strengths And Weaknesses:**

Strengths
1)The focus on LiDAR-based cross-view place recognition is innovative and addresses an important research gap.
2)The idea of aggregating spatio-temporal information and inferring probability density for position refinement is both interesting and theoretically sound.
3)The experiments show clear and significant improvements over prior methods, validating the effectiveness of the proposed approach.

Weaknesses:
1)The key to the proposed method is particle aggregation. However, section 3.4 is hard to follow. The usage of mathematical notations can be improved.
2)The collected datasets only cover two cities.
3)Missing metadata information of satellite images, such as capturing time, locations, and sensors.
4)Format issues noticed. Table 5 appears before Table 4.

---

> ### Author Rebuttal · Authors · 2025-07-31
>
> # Q1 and Q5: Sec. 4.3 is difficult to understand.
>
> Thank you for your suggestion. We acknowledge that the current presentation of Section 4.3 can be improved in terms of clarity and the use of mathematical notations.
>
> Section 4.3 introduces the STPE algorithm, which is designed to alleviate the unreliability of individual query retrieval results by leveraging information from a sequence of past queries.
>
> We utilize a sequence of past queries. For each query, we treat its top-K retrieval results as particles, where the state of each particle is defined by its x and y coordinates. Eq. 3 illustrates that, if the motion model is known, each particle can be propagated forward in time, with added random noise reflecting uncertainty. In practice, the motion model at each time step is directly obtained using FastGICP combined with magnetometer-based heading correction.
>
> The original intention of Eq. 4 and Eq. 5 was to express the probabilistic distribution of a particle's state after motion in a single time step. However, these equations may hinder understanding rather than clarify it, so we propose to remove them.
>
> Eq. 6 models the initial distribution of particles at a single past time step, which we assume follows a Gaussian Mixture Model (GMM). Since the x and y dimensions are independent, we use a diagonal covariance matrix (this step was omitted for brevity).
>
> Eq. 7, Eq. 8, and Eq. 9 describe how the key parameters of the GMM are estimated. Eq. 10 models the distribution of particles propagated from a single past time step to the current time step. Finally, Eq. 11 aggregates the distributions from multiple past time steps to form the probability density function at the current position. Eq. 12 then computes the probability score for each candidate remote sensing submap, which is used to re-rank the retrieval results for the current query.
>
> In the camera-ready version, we will carefully revise this section to provide a clearer explanation of the particle aggregation process. We will also include intermediate steps and intuitive descriptions to better guide readers through the derivation, ensuring that this part is more accessible and readable in the final version.
>
> # Q2 and Q6: The datasets only covers two cities.
>
> **Table 1 : Zero-shot performance comparison on Kitti-360 Seq. 00 and Oxford-L\&RSI .**
> _Recall@1_ and _Recall@10_ (%)↑ are reported.
> ||Kitti-360|Oxford|Oxford|
> |---|--|--|--|
> | Methods |Seq. 00 |11-14-02-26|14-12-05-52|
> | GeoDTR| 0.72 / 6.07|0.47/3.18 |0.48/3.71|
> | Sample4Geo | 5.84 / 23.64|7.13/20.68| 7.51/21.75|
> | FRGeo  | 2.87 / 14.10  |1.62/7.34 |1.58/7.86|
> | L2RSI (w/o STPE) | 6.46 / 24.67|10.19/30.23 |11.25/30.58|
> | L2RSI (w. STPE) | **41.20 / 67.84**|**42.29/59.41**| **43.77/62.76**|
>
> We appreciate the concern of the reviewer. We are also willing to perform broader evaluations. However, existing mainstream localization datasets do not provide corresponding high-resolution remote sensing images. Due to time constraints during the rebuttal period, we selected Sequence 00 from the KITTI-360 and manually constructed a new zero-shot test set by equipping it with corresponding satellite imagery. The data processing is consistent with that of XA-L\&RSI and Oxford-L\&RSI. The new test set contains **$1532$** Lidar submaps, **$2747$** remote sensing submaps, and remote sensing images covering an area of **$2.28km^2$**. As shown in **Table 1**, our method still achieves the best performance on this new benchmark. We will provide additional results trained and evaluated on KITTI-360 in the camera-ready version.
>
> # Q2 and Q6: Dependence segmentation models
>
> **Table 2: Ablation study about different semantic estimators.**
>
> _Recall@1_ and _Recall@10_ (%)↑ are reported.
> | Semantic Estimator | Methods             | Test Set-S        | Test Set-M        | Test Set-L        | Test Set-G        |
> | -- |-- | -- | -- | -- |-- |
> | HSSC | L2RSI (w/o STPE) | 29.62 / 69.22     | 21.11 / 57.04     | 18.91 / 53.57     | 17.75 / 48.32     |
> |                    | L2RSI (w. STPE) | 86.27 / 95.90     | 84.73 / 91.92     | 79.75 / 85.71     | 76.99 / 81.40     |
> | SphereFormer       | L2RSI (w/o STPE)    | 30.05 / 68.56     | 23.90 / 61.95     | 21.93 / 57.42     | 20.07 / 52.55     |
> |                    | L2RSI (w. STPE) | **88.93 / 92.13** | **87.95 / 91.64** | **85.49 / 90.65** | **83.27 / 87.82** |
>
> Thanks for your question. To evaluate the robustness of our method, we replaced SphereFormer with HSSC [1], and observed only a minor drop in performance. This indicates that our framework is not sensitive to the specific choice of segmentation model, as shown in **Table 2**.
>
> **[1]** Chen, S., Yang, B., Xia, Y., et al. *Bridging LiDAR gaps: A multi-LiDARs domain adaptation dataset for 3D semantic segmentation*. In *Proc. of IJCAI*, 2024, pp. 650–658.
>
> # Q3: Missing metadata information of satellite images
>
> We thank the reviewer for pointing this out. For XA-L\&RSI, remote sensing image data was captured in 2022 and 2023, covering the Xiang'an District of Xiamen, China (approx. $24^\circ32^\prime N$–$24^\circ41^\prime N$, $118^\circ10^\prime E$–$118^\circ20^\prime E$). For Oxford-L\&RSI, remote sensing image data was captured in 2019, covering the Xiang'an District of Xiamen, China (approx. $51^\circ44^\prime N$–$51^\circ45^\prime N$, $1^\circ14^\prime W$–$1^\circ16^\prime W$). All the remote sensing image data was collected by the Beijing-3 satellite, using sensors BJ3N1, BJ3N, BJ3N2, and BJ3A1.
>
> # Q4: Format issues noticed.
>
> Thank you for your careful review. We will thoroughly check the entire paper for formatting and spelling issues, and make the necessary corrections in the camera-ready version.

---

> > ### Comment · Reviewer_z1Y6 · 2025-08-05
> >
> > Thanks for author's response. Most of my concerns have been addressed. I would be happy to raise my score.

---

> ### Author Response · Authors · 2025-08-05
>
> Thank you for your kind decision to raise your score and for the encouraging feedback. We deeply appreciate your recognition of the progress we've made.
>
> Your suggestions regarding enhancing the clarity and presentation of our paper are truly valuable, and we agree that implementing them will improve both the paper's quality and its readability.
>
> As we incorporate these improvements, please let us know if any other aspects concerning clarity or other points require further attention.​​ We remain open to further discussions to ensure that the highest standards are achieved.
>
> Thank you once again for your valuable time and feedback.

---

### Official Review · Reviewer_hHQY · 2025-07-01

**Clarity:** 3
**Significance:** 3
**Originality:** 3
**Rating:** 4
**Confidence:** 2

**Summary:**

This paper tackles the problem of place recognition in large-scale urban environments using LiDAR point clouds without relying on a pre-built 3D map. Traditional methods either assume the availability of costly 3D maps or perform retrieval only within limited areas, which hinders scalability. To overcome these limitations, the authors propose L2RSI, a contrastive-learning framework that aligns LiDAR submaps with high-resolution remote sensing imagery in a shared semantic space. They further introduce STPE (Spatial-Temporal Particle Estimation), which models the spatio-temporal information of consecutive queries as a particle set to compensate for the instability of single-query predictions. Experiments on the XA-L&RSI and Oxford-L&RSI datasets demonstrate that validating both its effectiveness and its generalization capability.

**Questions:**

1. Will you publicly release the XA-L&RSI and Oxford-L&RSI datasets?

2. Are there alternative strategies that could achieve comparable performance without using ego-motion information?

3. How was end-to-end latency managed during real-time robotic operation, and what was the total pipeline delay?

**Ethical Concerns:**

["NO or VERY MINOR ethics concerns only"]

**Final Justification:**

My primary concerns regarding STPE overhead, dataset availability, and real-time latency have been fully resolved. I therefore maintain Borderline Accept.

**Limitations:**

Yes.

**Paper Formatting Concerns:**

None.

**Quality:**

3

**Strengths And Weaknesses:**

Strengths:


1.The paper provides a clear problem definition and strong motivation, and the methodology is presented in a logical, easy-to-follow manner.

2. It convincingly demonstrates the effectiveness of combining semantic contrastive learning with STPE for cross-view, cross-modal place recognition over a large (100 km²) urban area.

3. Thorough evaluations on multiple datasets and retrieval ranges showcase the robustness and generalization of the proposed approach.


Weaknesses:


1. There is insufficient discussion on how to minimize the additional overhead introduced by the STPE module for real-time applications.

2. The reliance on ego-motion measurements and magnetometer data raises concerns about applicability in scenarios where these sensors are unavailable.

---

> ### Author Rebuttal · Authors · 2025-07-31
>
> # Q1: How to reduce STPE overhead for real-time use?
>
> Thank you for the insightful question. In practice, since our L2RSI framework incorporates the efficient FastGICP algorithm, the STPE module does not need to be activated for every query. Instead, it can be triggered periodically for high-precision place recognition. Between two STPE activations, relative poses can be propagated using FastGICP. This trigger-based design significantly reduces the computational overhead introduced by STPE in real-time applications.
>
> # Q2 and Q4: Dependence on magnetometer data.
>
> **Table1 : Ablation study about effectiveness of components. (Table 4 in the main paper)**
> Results include _Recall@1_ and _Recall@30_ (%)↑ on **Test Set-S** and **11-14-02-26**.
> **Sem.**: Semantic segmentation. **STPE**: Spatial-Temporal Particle Estimation. **Orient.**: Orientation.
> The best results are in **bold**.
>
> | Sem. | STPE | Orient. | Test Set-S | 11-14-02-26|
> |:--:|:--:|:--:|:--:|:--:|
> || ✔ | ✔| 50.43 / 67.90  | 8.13 / 25.42  |
> | ✔    || ✔       | 30.05 / 85.50  | 10.19 / 45.90 |
> | ✔    | ✔    || 54.37 / 76.14| 4.92 / 21.24|
> | ✔| ✔ | ✔ | **88.93 / 95.94** | **42.29 / 70.15**|
>
> As noted in our Limitations section (A.1 in the Appendix), the proposed framework relies on magnetometer data to provide a coarse orientation prior. This is crucial because, at the city scale (covering an area of $100 km^2$), LiDAR-to-remote-sensing-image retrieval is highly challenging. Even a rough orientation estimate significantly facilitates the learning of cross-view and cross-modal global descriptors. As shown in Table 1 (third row),  removing this directional input significantly degrades the performance of place recognition. In future work, we plan to address this limitation by exploring orientation-invariant or self-supervised alternatives.
>
> # Q3: Datasets open source.
>
> Thank you for your interest in our dataset. We have already obtained the necessary open-source licenses for remote sensing imagery. The XA-L&RSI and Oxford-L&RSI datasets will be released to the research community upon the official publication of the paper.
>
> # Q5: End-to-end latency during real-time robotic operation.
>
> Thanks for your valuable question. The total inference latency of the proposed L2RSI pipeline is approximately **$290ms$**, which includes around **$200ms$** for point cloud semantic segmentation, **$50ms$** for generating the semantic BEV image, and **$40ms$** for retrieval and STPE refinement per query. It is worth noting that our current implementation focuses on offline evaluation to validate the core contributions, and has not yet been optimized for deployment. In practical applications, the system can adopt a trigger-based design, where high-accuracy place recognition is performed periodically, which is sufficient to meet the requirements of typical urban autonomous localization tasks.

---

> > ### Comment · Reviewer_hHQY · 2025-08-05
> > **Thank you for the thorough rebuttal**
> >
> > The authors have addressed the majority of my concerns regarding STPE overhead, sensor dependencies, and real-time latency, so I will maintain my recommendation of Borderline Accept.

---

> ### Author Response · Authors · 2025-08-05
>
> Thank you for the time and effort you have dedicated to reviewing our work. We truly appreciate your valuable feedback and thoughtful suggestions. Should you have any concerns or points that require further attention, we would be happy to continue the discussion and address them to ensure the highest standards are achieved.

---

> > ### Comment · Reviewer_hHQY · 2025-08-06
> > **Concerns Regarding Real-Time Latency**
> >
> > The core contribution of this paper is vehicle localization, which inherently demands real-time performance. Reporting a 290 ms per-cycle latency for localization alone—measured on a high-end desktop GPU (RTX 3090)—reveals a clear limitation of this work. Autonomous vehicles must run not only localization but also detection, voxel-occupancy prediction, tracking, and other perception tasks in parallel; dedicating 290 ms solely to localization is excessive.
> >
> > While I acknowledge that the current implementation serves as an offline validation of the core idea, practical deployment will require explicit discussion of model lightweighting and efficiency optimizations. Given that LiDAR sensors typically operate at 10–20 Hz, executing this method online—particularly on more constrained, embedded GPU platforms—will demand a more refined design and further acceleration strategies.
> >
> > Although this computational overhead does not justify outright rejection, I view it as a moderate limitation and therefore maintain my recommendation of Borderline Accept, as I believe it is unlikely to be fully resolved during the rebuttal phase.

---

> ### Author Response · Authors · 2025-08-06
>
> We appreciate the reviewer’s thoughtful comments regarding runtime efficiency and its implications for real-world deployment.
>
> As mentioned, our current reported latency of ~290 ms per query was obtained on an RTX 3090 GPU using an unoptimized Python implementation. This version includes substantial debug functionality, logging, and data dumping routines, which are not representative of a production system. Therefore, the reported time reflects a conservative upper bound.
>
> In practice, we acknowledge that achieving real-time performance is critical for on-vehicle deployment. To this end, we foresee multiple engineering directions to significantly reduce latency, including:
>
> 1. Codebase Refactoring and Compilation: Our current implementation is entirely in Python. Migrating core modules to C++ or leveraging PyTorch’s JIT compilation can substantially reduce overhead.
> 2. Triggered Execution Mode: In practical systems, Fast-GICP runs in real-time to provide relative poses between consecutive point cloud scans. The proposed L2RSI module only needs to operate in a triggered manner, intermittently providing accurate global localization results to align the relative poses from Fast-GICP to a global coordinate frame. As shown in Table 5 of the main paper, the STPE algorithm can function effectively even when the sequence sampling rate is reduced to 30%. Therefore, under the triggered execution mode, the point cloud semantic segmentation, BEV image generation, and retrieval components of L2RSI operate at a low frequency. The STPE algorithm is only executed upon trigger events.
>
> **Table 5: Ablation study about the sampling rate $\lambda$.**
> *Recall@1 (%)* on **Test Set-S** is reported. The runtime (ms) here only involves **STPE**. The best result is in **bold**.
>
> |**Test Set-S**|**10%**|**20%**|**30%**|**40%**|**50%**|**60%**|**70%**|**80%**|**90%**|**100%**|
> |--|--|--|--|--|--|--|--|--|--|--|
> | **Recall@1(%)↑**| 85.98  | 88.07  | **88.93** | 88.44  | 88.44  | 88.07  | 88.07  | 87.82  | 88.07  | 88.44  |
> | **Runtime(ms)↓**| **24.1** | 29.4|31.7| 34.7| 38.0| 40.2| 44.3| 47.5|52.2| 55.4|
>
> 3. We fully agree with you that reaching practical deployment speeds—especially for embedded platforms—will require substantial engineering effort. Our long-term goal is to achieve a 10 Hz localization rate. These optimizations will be continuously incorporated and tracked via our GitHub repository.
>
> Thank you once again for your valuable feedback and thoughtful comments.

---

### Note · Authors · 2025-08-12

We sincerely thank all reviewers for their constructive feedback and the valuable discussions during the rebuttal phase. These exchanges have helped us clarify our arguments and refine technical details, which we believe have significantly strengthened the quality of the paper. They have also provided us with a clearer direction for future work.

We would like to reiterate the key contributions of our work:
- We are the first to address the challenge of large-scale (over 100 km²) urban cross-view LiDAR-based place recognition with high-resolution remote sensing imagery.
- The integration of Spatial-Temporal Particle Estimation (STPE) provides a principled and innovative solution by exploiting both spatial and temporal continuity in sequential queries, which significantly improves retrieval accuracy (Recall@1 increased from 20.07% to 83.27%).

If accepted, we will release the dataset and maintain a public GitHub repository to ensure reproducibility of the results presented in the paper, as well as to support continuous improvements and updates.

---

### Decision · Program_Chairs · 2025-09-17

**Decision:**

Accept (poster)

**Comment:**

This paper addresses the challenge of LiDAR-based place recognition. It introduces a new dataset, XA-L&RSI, which contains approximately 110,000 remote sensing submaps and 13,000 LiDAR point cloud submaps captured in urban scenes. It further proposes a method, L2RSI, for cross-view LiDAR-based place recognition.

The paper initially received mixed scores (2× borderline accept, 1× borderline reject, 1× reject). Reviewers highlighted several strengths:
- The motivation is clear, the task is important, and the dataset is valuable;
- The proposed method is simple, effective, and technically sound;
- The evaluation is thorough.

Concerns were raised about the method’s novelty, paper clarity, completeness of related work, and the dataset license. The authors provided a strong rebuttal that effectively addressed these issues, resulting in all positive final scores (2× borderline accept, 2× accept). The AC concurs with this recommendation.